# Transform-Enabled Detection of Backdoor Attacks in Deep Convolutional Neural Networks

## Abstract

Deep Neural Networks (DNNs) have been widely deployed in a range of safety-critical applications. Recent work has illustrated their vulnerability to malicious backdoor attacks, which lead to DNN malfunction when a specific backdoor trigger is applied to the DNN input image. These backdoors cause uncharacteristic behavior in DNN hidden layers, causing the DNN to misclassify the input image. In this work we present Transform-Enabled Detection of Attacks (*TESDA*), a novel algorithm for on-line detection of uncharacteristic behavior in DNN hidden layers indicative of a backdoor. We leverage the training-dataset distributions of reduced-dimension transforms of deep features in a *backdoored* DNN to rapidly detect malicious behavior, using theoretically grounded methods with bounded false alarm rates. We verify that TESDA is able to achieve state-of-the-art detection with very low latency on a variety of attacks, datasets and network backbones. Further ablations show that only a small proportion of DNN training data is needed for TESDA to fit an attack detector to the backdoored network.

## 1 Introduction

The increasingly widespread deployment of Deep Neural Networks (DNNs) has led to the use of third-party providers to train and provision these DNNs at scale (Ribeiro et al., 2015). DNNs have been used in a range of safety critical applications such as autonomous driving (Grigorescu et al., 2020) and medical robotics (Akay & Hess, 2019), where reliability and security of the DNN system is of paramount concern. In such applications, access to proprietary datasets may be granted to a third-party provider for model training. However, the use of third-party providers has been shown in prior work to make DNNs vulnerable to security threats (Chen & Koushanfar, 2023). As such, securing models against such threats has become of prime interest to the community.

One model for attacks on DNNs is the adversarial attack, wherein imperceptible perturbations are added to the DNN inputs in order to force misprediction (Chakraborty et al., 2021) often using time-consuming optimization-based methods. Another paradigm which is the focus of this work is that of *backdoor* attacks, which inject a backdoor into the DNN model during the training process to force misprediction when presented with a *trigger* (a unique stimulus superimposed on or applied to the model inputs) (Chen & Koushanfar, 2023). These backdoored DNNs thus consistently mispredict in the presence of the trigger, while behaving nominally in its absence. Such backdoors may be injected during model training in a variety of ways, at different stages of the model training process. Poisoned data may be used to train the model to mispredict in the presence of the trigger, as in (Gu et al., 2019; Nguyen & Tran, 2021). Alteration of the training process itself may also be used to insert a backdoor into the network (Liu et al., 2020c). After training, bitflips may be injected into the network or weights may be perturbed to force misprediction in the presence of a trigger (Rakin et al., 2020). All of these methods can be used by an attacker who has access to the entire model training pipeline, as is the case with third-party training providers. In this paper, we assume that the defender has access to training data and to the final trained (backdoored) model and *no* access to the training pipeline, as is the case when using proprietary data and a third-party MLaaS provider.

The backdoor triggers placed in the deep learning model cause sharp changes in neuron behavior on being presented with the trigger pattern that result in misprediction. Prior art has leveraged channel Lipschitzness bounds on layer outputs to detect changes in behavior (Zheng et al., 2022a) in layers most affected by backdoor insertion. Related prior art has analyzed adversarial perturbations in network inputs (Cantareira et al., 2021) to find activations that diverge significantly in behavior under adversarial input perturbations. We leverage the observation that the forced misprediction caused by backdoor attacks can be easily detected using statistical tests of transformed, reduced-dimension deep feature (intermediate layer output) distributions due to this 'diverging' behavior under forced model misprediction.

Defenses against backdoor attacks can be broadly categorized into offline methods that modify or retrain the compromised model, and online methods that detect malicious inputs during inference. Unlike offline defense methods that require model retraining or modification, online detection systems enable real-time protection during inference while preserving the original model's functionality. This capability is essential for deployed systems where taking models offline for remediation is impractical or where individual suspicious inputs must be flagged for human review. This paper presents TESDA, a modular low-cost approach to detection of backdoor attacks in lightweight edge-based deep convolutional neural networks that leverages distributions of reduced-dimension representations of deep features of neuron activations in clean training data for detection of reduced-dimension transformed deep features indicative of a backdoor attack (uncharacteristic, 'diverging' deep features (intermediate layer outputs) within the DNN). TESDA moreover does not require access to backdoor trigger information and is trigger-agnostic, detecting potential threats based on the behavior of the backdoored network on clean data. We provide theoretical bounds on false alarm rates in the TESDA algorithm, leverage its modular structure to validate our methodology on different deep feature transforms, varying deep feature distribution dimensions and varying amounts of training data to build the deep feature distribution. We further show that TESDA provides real-time detection capability with low overhead.

The following section discusses prior related work in the field, and Section 3 provides an overview of the TESDA algorithm and its modular blocks. Section 4 discusses algorithmic details and establishes bounds on false alarm rates for outlier (backdoor attack) detection. Section 5 presents the experimental setup, baselines and ablation configurations for TESDA, while Section 6 discusses the results of these experiments. We discuss specific findings and limitations of TESDA and angles of future research in Section 7, and conclude in Section 8.

## 2 Prior Work

We begin by presenting a brief overview of work in deep learning backdoor attacks (also referred to as *Neural Trojans*), and then discuss prior work on defense and detection of these attacks before presenting the key contributions of our proposed approach.

### 2.1 Backdoor Attacks

As mentioned in Section 1, a backdoor (Trojan) attack forces misprediction by the DNN when a trigger is superimposed its input data. In this work we focus on image inputs to classifier networks as the backdoor attacks' target. We focus on training-time data poisoning based backdoor attacks in this work, as opposed to post-training bitflip-based attacks such as (Rakin et al., 2020) that induce hardware malfunction. These attacks, while addressable using our approach, are addressed in a more cost-effective fashion at the hardware level by detecting the backdoor-induced hardware malfunction (Liu et al., 2020a).

These trigger patterns can be *fixed* (one trigger pattern for all inputs), with the DNN trained to mispredict on being presented with the set trigger pattern, as shown in (Liu et al., 2018b). A fixed trigger can also be 'blended' with the input image, providing a larger but less perceptible trigger pattern wth no size constraint (Chen et al., 2017). A more stealthy fixed 'optical' Trojan trigger was shown in (Boloor et al., 2021), mimicking a physically realizable lens with a trigger pattern embedded in it to force misprediction. Label-consistent poisoned inputs have also been used to reduce the chance of fixed triggers being detected through visual inspection (Turner et al., 2019). Unlike the previous fixed trigger Trojans, the Blind attack (Bagdasaryan &

Shmatikov, 2021) compromises the loss computations to force misprediction using fixed triggers, as well as semantic or transform-based triggers.

In contrast, *Input-dependent triggers* use a trigger function applied to the input image for trigger generation, generating a unique trigger pattern for each input. These dynamic backdoors were first shown in (Salem et al., 2022), using a conditional Backdoor generation Network (c-BaN). The input-aware backdoor (Nguyen & Tran, 2020) accomplished this using a cross-trigger loss to ensure that triggers remain sample-specific. Trigger generation was done using an encoder-decoder network that was trained using diversity loss. The Single-Sample Based Attack (SSBA) (Li et al., 2021c) instead used sample-specific additive noise as a trigger by encoding attacker-specified strings into benign images using an encoder-decoder network.

More subtle *transform-based triggers* use image transforms as trigger functions, rather than pixel-level changes. ReFool (Liu et al., 2020b) uses mathematical models of reflection to plant imperceptible *input-dependent* triggers (as opposed to fixed triggers) to force misprediction. WaNet moved away from local image transforms such as ReFool to imperceptible warping transforms on the entire image (Nguyen & Tran, 2021). BPP (Wang et al., 2022b) uses small input-dependent quantization of color channels in input images to force misprediction, using adversarial training and contrastive learning for trigger injection. LIRA (Doan et al., 2021b) frames imperceptible trigger generation as a nonconvex constrained optimization solved using stochastic methods. The trigger function is thus a conditional generative function (input-dependent) formulated as a nondeterministic mapping (pixel-level), unlike WaNet and BPP. Later work (Doan et al., 2021a) uses a similar framing but uses an Wasserstein distance as a constraint on the optimization. Narcissus (Zeng et al., 2023) optimizes trigger patterns to point to the inside of the target class, using class-oriented trigger functions. It uses open-source Public Out-Of-Distribution data to make the attack, thereby ensuring practical applicability.

## 2.2 Backdoor Detection and Defense

We define backdoor *detection* methods as those that identify poisoned inputs or data, or identify behavior characteristic of misprediction caused by a Neural Trojan. This can be *offline*, to separate poisoned from clean data, or *online*, to defend a potentially compromised network during inference. Backdoor *defenses* are methods that remove Trojan triggers (if *online*), or reverse Trojan trigger injection to cleanse the network (usually *offline*). In this taxonomy, TESDA is an *online detector* of backdoor attacks.

*Offline detection* methods such as (Liu et al., 2017) often leverage the distributional differences between clean and poisoned data to detect poisoned data. A simple method proposed in (Chen et al., 2018) clusters the activations of the last hidden layer after PCA or ICA, using k-means to form a clean and poisoned data cluster. ASSET (Pan et al., 2023) more recently uses a two step optimization to separate poisoned from clean samples in training data. It minimizes the loss on a clean base set, then maximizes it on the training dataset, using a loss threshold to detect poisoned samples. SPECTRE (Hayase et al., 2021) strips poisoned data out of the training dataset by estimating the mean and covariance of clean training data using robust covariance estimation, and whitening the combined data using the estimated statistics. Activation gradients of poisoned training data can also be used to identify it (Yuan et al., 2024a), using the fact that the gradient circular distribution of the target class for Trojan misprediction is more dispersed than the clean class. Provenance-based data filtering proposed in (Baracaldo et al., 2018) filters input data using provenance features that describe the origin and lineage of the data for the partially trusted case, using a classifier to cluster and separate poisoned and clean data. Spectral signature based detection on training data (Tran et al., 2018) is done based on the L2 norm of the top singular vector of each class, comparing the clean (training) class spectrum with the poisoned training spectrum. Feature Consistency to Transforms (FCT) (Chen et al., 2022b) metrics have also been used to separate clean, poisoned and suspicious data. Retraining on clean data and unlearning the poisoned samples is used for cleansing the DNN. Self-supervised learning methods (Chen et al., 2022a) have also been used to decouple the backdoor features from the target class and filter out high-credibility samples via label-noise learning.

Universal Litmus Pattern inputs to DNNs can also be used to detect a Trojan without training data statistics or clean data (Kolouri et al., 2020). This approach requires a classifier to be used for the binary poisoned/not poisoned task and needs training data consisting of poisoned and clean models. Similarly, (Xu et al., 2021)

train a meta-classifier to identify whether or not the model being studied is Trojaned. This trigger reverse-engineering can also use adversarial perturbations on every class-pair, as in (Xiang et al., 2020) to detect the presence of a Trojan. FreeEagle (Fu et al., 2023) is a data-free offline backdoor detection method that forward-propagates intermediate feature representations to maximize the posterior probability of each class, detecting a Trojan if the features for other classes have an abnormally high posterior probability on a single given class.

*Online detection methods* at inference time are used to detect or defend against Trojaned inputs or behavior indicative of Trojan-induced misprediction in a compromised DNN. STRIP (Gao et al., 2019) perturbs input images by blending with other training images from different classes, recording entropy of the network logits for each perturbation. An abnormally low entropy value indicates the Trojan trigger forcing identical behavior (misclassification). TeCo (Liu et al., 2023) builds on this by using image corruptions to perturb the inputs, and uses a classifier corruption robustness metric to detect Trojaned inputs - inputs with a Trojan trigger are more robust to image corruptions. Both TeCo and STRIP require high overhead (75-100X the DNN latency), since they run DNN inference not just on the input image but on each perturbed image as well. More recent work (Tang et al., 2021) requires access to a subset of poisoned data to build a discriminating hyperplane between classes, and uses a likelihood ratio test to estimate whether an input sample is poisoned. CLEANN (Javaheripi et al., 2020) performs GMM-driven outlier detection of the error in sparse reconstructions of inputs from DNN feature maps, and orthogonal matching pursuit to find the correct class from a Trojaned image. To be on-line, CLEANN uses dedicated hardware due to expensive iterative computations, and is the single on-line defense method examined thus far.

*Online detection* methods such as NeuronInspect use features extracted from expensive Grad-CAM heatmaps to build a composite feature for outlier (Trojaned input) detection (Huang et al., 2019). SentiNet (Chou et al., 2020) likewise uses Grad-CAM features to generate class proposals from input segmentation and the classifier output to detect Trojaned inputs to a DNN, checking to see if certain segmentation regions disproportionately impact classification (indicating Trojan trigger presence). This may not work for nonlocal triggers (such as Blended triggers or WaNet), and requires repeated DNN inference for each input image as well as expensive Grad-CAM. RAID (Fu et al., 2022) instead uses features extracted by the feature extraction backbone of a CNN and a novelty detector (on the PCA of the features) combined with a shallow neural network whose inferences are compared with that of the network itself to decide if incoming data is Trojaned or not. Similar use of an auxiliary DNN is seen in (Subedar et al., 2019), fitting class-conditional distributions to the features of the DNN after training. At inference, log-likelihood scores of test feature samples w.r.t. these distributions are collected. Clean samples have high likelihood, while Trojaned ones have low likelihood. Using noisy SGD to learn differentially private NNs is shown in (Du et al., 2019) to accomplish uniform asymptotic empirical risk minimization. This is then used for enhanced outlier detection in autoencoders, detecting poisoned samples fed to a backdoored neural network.

*Offline defenses* against Neural Trojans span a wider range of techniques. Neural Cleanse (Wang et al., 2019) reverse-engineers triggers in compromised DNNs by finding the minimal perturbation needed for each class to be Trojaned, using an outlier detector to find a Trojan (if the perturbation required to mispredict to a target class is significantly different). Triggers are removed by pruning or unlearning. Gangsweep (Zhu et al., 2020) used a GAN to detect and remove neural trojan backdoors in a DNN via generation of a perturbation mask. Retraining with reconstructed triggers and modified loss to reduce trigger sensitivity restores clean performance. Anti-Backdoor Learning (Li et al., 2021a) used a two-stage gradient ascent mechanism to isolate backdoor samples and break the correlation between backdoor samples and the target class. Backdoor unlearning has also been done by learning implicit hypergradient (Zeng et al., 2021), formulating a minimax optimization problem for Trojan defense. Generative Distribution Modeling (Qiao et al., 2019) uses max-entropy staircase approximation for sampling-free high-dimensional generative modeling to recover the trigger distribution. Pixel-wise adversarial perturbation to directly reverse-engineer Trojan triggers and data-free trigger reverse-engineering using these perturbations has been explored in (Wang et al., 2020).

Pruning of neurons responsive to the Trojan trigger is a common offline defense, starting with Fine-Pruning (Liu et al., 2018a), which alternated between pruning and fine-tuning as a defense against backdoors. Entropy Pruning (Zheng et al., 2022b) and MBNS (Zheng et al., 2022b) use differential entropy of clean and poisoned data distributions or distributions of minibatch norm statistics respectively to prune neurons that contribute

to Trojaned behavior, cleansing the network. ANP (Wu & Wang, 2021) instead uses adversarial perturbations to find such prunable neurons. Neural Attention Distillation (Li et al., 2021b) instead aligns neurons more responsive to the trigger with those that are benign. CLP more recently (Zheng et al., 2022a) uses a channel Lipschitzness threshold to identify Trojaned neurons for data-free pruning.

## 2.3 Key Contributions

This paper presents TESDA, an online detector for backdoor attacks on image classification Convolutional Neural Networks. We make the following assumptions w.r.t. the attack scenario:

- The attacker has full access to training and a copy of the training dataset, and provides a trained and backdoored network to the defender. The defender has no access to the training process.

- The defender has access to a portion or the whole of the clean training dataset, as is often the case w.r.t. proprietary data being used for ML-as-a-service.

- The defender has no access to poisoned data, backdoors or knowledge of the trigger size, shape, target class or trigger position. The defender can make no assumptions as to the trigger, and TESDA therefore is trigger-agnostic.

- The defender requires an on-line defense with minimal latency and compute overhead to be deployed with the DNN at inference, as opposed to offline model remediation approaches like fine-tuning.

The need for minimal overhead and latency restrict the use of techniques such as those in (Huang et al., 2019) and (Chou et al., 2020) that use expensive Grad-CAM and auxiliary DNNs. The need for repeated DNN forward passes to detect backdoors in STRIP and TeCo raises the latency by $O(75)$ (Liu et al., 2023) to $O(100)$ (Gao et al., 2019). Auxiliary DNNs for detecting attacks in RAID (Fu et al., 2022) and (Subedar et al., 2019) likewise impose high overhead costs. (Tang et al., 2021) requires access to poisoned data, an assumption we do not make. Lastly, while (Javaheripi et al., 2020) is real-time *with* specialized hardware, we do not assume that the deployment scenario always has access to such hardware.

However, the need for secure inference in safety critical scenarios requires high detection coverage similar to offline defense methods such as (Zheng et al., 2022b) or detection methods such as (Tran et al., 2018).

To address this problem, the key contributions of our work are:

- We provide a low-latency, online Neural Trojan detector, outperforming prior art.

- Our detector is seen to perform at state of the art levels for Neural Trojan detection.

- Our detector is modular, allowing substitution of components and design choices, which we cover in later sections. This allows redesign and adaptability to different use cases.

- We provide theoretical bounds on false alarm rates and utilize interpretable, well-understood methods such as Principal Component Analysis (PCA) and Minimum Covariance Determinant (MCD) outlier detection.

We thus leverage reduced-dimension low-overhead transforms of deep features of the neural network for theoretically grounded outlier detection. We also do not require access to all deep features in the network, and show in this work that TESDA can make use of as few as four layers' deep feature vectors in Preactresnet-18 for Trojan detection. The number of layers' deep features used for TESDA is a design parameter that can be varied to increase sensitivity at the cost of computational overhead. Prior use of PCA methods such as (Tran et al., 2018) did not leverage behavioral changes in deep features or required access to poisoned data (Chen et al., 2018), which we do not.

Finally, we note that while fine-tuning the final layers using clean data is a potential alternative defense(Zhu et al., 2023), TESDA addresses fundamentally different requirements for real-world deployment. Fine-tuning

requires taking the model offline, careful hyperparameter selection, and permanently modifies the model weights, potentially degrading performance on edge cases not represented in the limited clean data. In contrast, TESDA provides online protection during inference without modifying the original model. Furthermore, fine-tuning addresses the backdoor at the model level but cannot detect individual poisoned inputs during deployment, whereas TESDA enables real-time flagging of suspicious inputs for human review or rejection. The two approaches are thus complementary rather than competing solutions, with TESDA specifically designed for scenarios requiring continuous operation and real-time threat detection.

## 3 Approach Overview

### 3.1 Problem Statement

The Neural Trojan attack problem has been described mathematically by Wang et al. (2022a), to generate an input transform that results in a high probability of misprediction to a target class. The prediction process in the deep neural network (DNN) can be represented as:

$$g(x) = f_N \circ f_{N-1} \circ \cdots \circ f_1(x) \tag{1}$$

where each function $f_i$ represents computations at that layer of the DNN (total $N$ layers), $\circ$ represents the function composition created by consecutive layers acting on one another and $x$ is the input to the network drawn from the input space $\mathcal{X}$ ($x \in \mathcal{X}$). The prediction function $g(.)$ created by executing these DNN layers that map the input $x$ to a label $y$ in the label space $Y$.

The misprediction process induced by the neural trojan can therefore be described as creating a prediction function $\hat{g}(.)$, which through data poisoning maximizes misprediction to a target class $y^t$ when a backdoor trigger function $B(.)$ is applied to the input $x \in \mathcal{X}$. In the absence of the trigger function, $\hat{g}$ simply produces nominal, non-backdoored behavior. Mathematically, we describe this process as follows:

$$\max_{\hat{g}(.)} \left( P(\hat{g}(x^b) = y^t) - P(g(x) \neq \hat{g}(x)) \right), \quad \forall x \in \mathcal{X}, \text{ and } x^b = B(x) \tag{2}$$

where $\max_{\hat{g}(.)}$ indicates the maximum over all backdoor prediction functions $\hat{g}(.)$ of the probability that a backdoored input $x^b$ is misclassified to target class $y^t$. The created prediction function $\hat{g}(.)$ similarly minimizes the probability of misprediction for non-backdoored inputs in the second term ($P(g(x) \neq \hat{g}(x))$) of Equation 2. To detect this altered prediction function $\hat{g}(.)$ forcing misprediction when presented with a trigger, earlier work such as (Chen et al., 2018) has focused on clustering its outputs over a poisoned training dataset and detecting poisoned inputs (i.e., $x^b = B(x)$) using distribution distances. We take this one step further, building on observations of adversarial perturbation dynamics in (Cantareira et al., 2021), and frame the problem as outlier detection:

$$Tesda(x^b) = Outlier([T_i(f_i \circ f_{i-1} \circ ... \circ f_0(x^b)]_{i=0}^N) \tag{3}$$

where transforms $T_i(.)$ are applied to the $i$th layer output $[f_i \circ f_{i-1} \circ ... \circ f_1(x^b)]_{i=0}^N$ ($1 \leq i \leq N$), and the vector of transformed layer outputs is used for outlier detection based on the distance from the distribution of *clean* transformed layer outputs. We note that all layer outputs need not be used - a subset of layer outputs $\{i \in L | L \subset [1, 2, ...N]\}$ can also be used to reduce computation overhead. Since we make use of the distribution of clean transformed layer outputs (deep features), TESDA is trigger-agnostic - any backdoor trigger that causes a sufficiently uncharacteristic shift in transformed deep features will be detected.

The choice of transforms $T_i(.)$ is done to improve detection capability (capturing large behavioral changes in deep features shown in (Cantareira et al., 2021)) while also reducing compute overhead from processing high-dimensional deep feature outputs from multiple layers. These choices are discussed in the following section.

### 3.2 High-Level Flow

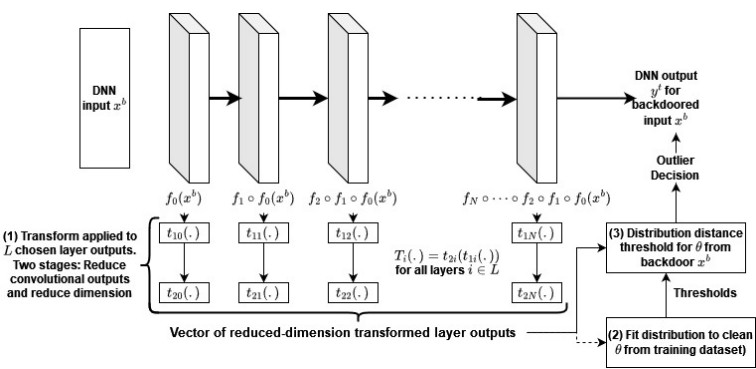

Figure 1 shows an overview of the TESDA backdoor detection process applied to a neural network. TESDA consists of three main steps applied to the DNN after it is trained and potentially backdoored:

*First (Step 1 of Fig. 1)*, TESDA applies a transform function to each of the layer outputs chosen for outlier detection, $T_i(f_i \circ f_{i-1} \circ ... \circ f_0(.))$, for a layer index $i$ out of the set of chosen layers $L$ ($\{i \in L | L \subset [1, 2, ...N]\}$). This transform consists of two steps, as seen in Fig. 1: (1) A function $t_{1i}(f_i(...))$ applied to outputs of convolutional layers (feature maps) to reduce each kernel output's dimension (e.g. taking the variance across the feature map; taking the feature map

Figure 1: An overview of the TESDA process for backdoor detection applied to a neural network. Deep features (layer outputs) consist of a vector of 2-D feature maps for a convolutional layer. They are transformed in two steps ($t_1i$ and $t_{2i}$) to low-dimensional outputs (Step 1), a clean dataset of these outputs is fitted to a distribution (Step 2) and concatenated to a vector for outlier detection (Step 3).

mean value); and (2) Use of a dimensionality reduction transform $t_{2i}(.)$ on the reduced layer outputs to yield a low-dimensional transformed output for each layer, so that $T_i(x) = t_{2i}(t_{1i}(.))$ in Equation 3. We have used the Principal Component Analysis (PCA) transform (Shlens, 2014) here and discarded all but a few chosen PCA coefficients to produce the output of each $T_i(.)$. The goal of this step is to highlight characteristics of data that are sensitive to shifts caused by $x^b$ while reducing the high-dimensional deep feature outputs to something tractable for on-line detection.

*Second (Step 2 in Fig. 1)*, over a subset of clean training data, TESDA fits a distribution $\mathcal{D}^C$ to the vector of transformed clean layer outputs $[T_i(f_i \circ f_{i-1} \circ ... \circ f_0(x)]_{i \in L}$. This is a computationally intensive step done before deployment of the DNN in the field.

*Third (Step 3 in Fig. 1)*, an outlier detection system is used to flag backdoored outputs. Since, per Equation 2, there is a large difference in behavior for $\hat{g}(.)$ when backdoored or clean, we use the distance of the vector $[T_i(f_i \circ f_{i-1} \circ ... \circ f_0(x^b)]_{i \in L}$ from the distribution of clean outputs $\mathcal{D}^C$ for outlier detection in the function $O(.)$ of Fig. 1. Since this detector is run for on-line detection, we require a fast, theoretically grounded algorithm. We have used a Gaussian elliptic envelope outlier detector (Rousseeuw & Driessen, 1999) here to fit a distribution to the vector of $T_i(.)$ and flag backdoored inputs based on distribution distance.

Each of these modules (in Steps 1, 2 and 3) can be filled by multiple methods, and we explain the design choices we have used here in the following section in detail.

## 4 Methods

### 4.1 Deep Feature Transformation

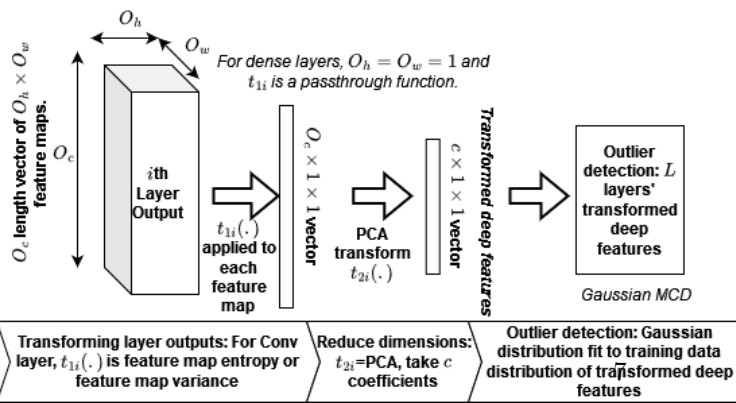

In this step (Step 1 of Fig. 1), the deep features of the $i$th layer in the network are subjected to a transform $t_{1i}(.)$ to highlight characteristics of data that are sensitive to shifts caused by a backdoor while reducing data dimension. Fig. 2 illustrates this for a convolutional layer, where the three-dimensional output tensor is reduced to a vector using a transform $t_{1i}$. For dense layers, we simply use $t_{1i}(x) = x$, a pass-through function,

Figure 2: The transform process to map a deep feature vector (layer output) to a low dimensional vector for outlier (backdoor) detection. As in Section

since the output of a dense layer for a single input is a vector already.

In Fig. 2, the goal is to reduce the $[O_c, O_h, O_w]$ dimension tensor of outputs, where $w$ denotes width, $h$ denotes height and $c$ denotes channel dimension. This is reduced in the first step of Fig. 2 to a vector of dimension $[O_c, 1]$, so that each feature map of dimension $[O_h, O_w]$ is reduced to a scalar. In this work we experiment with two transforms: (1) Feature map entropy (giving a vector of feature map entropies), calculated for each feature map in a convolutional layer output (Equation 4); and (2) The variance of each feature map (giving a vector of feature map variances) (Equation 5):

$$t_{1i}(x) = \sum_{k=1}^{O_h} \sum_{j=1}^{O_w} g(x_{k,j}) \log(g(x_{k,j})) \forall x \in X \tag{4}$$

$$t_{1i}(x) = Var(x) \forall x \in X \tag{5}$$

where $x$ is an $O_h \times O_w$ dimensional feature map, the $O_c$-dimensional vector of these feature maps output from the $i$th layer is $X$, the convolutional layer output and $g(x_{k,j}) = \frac{e^{x_{k,j}}}{\sum_{O_h, O_w} e^{x_{k,j}}}$ is the softmax function applied to the elements of the 2-D feature map $x$ in the $i$th layer.

The first transform (Equation 4) builds a distribution of reduced-dimension feature map entropy vector to detect changes in entropy similar to prior work (Gao et al., 2019). The second transform (Equation 5) builds a distribution of reduced-dimension feature map variance vectors to check for uncharacteristic features being highlighted in the deep features (indicative of a trigger applied to $x$). Ablations covering both transforms are found in Section 6. For brevity, in the succeeding sections we denote the application of $t_{1i}$ to each feature map in the layer output $X$ as $t_{1i}(X)$. For a dense layer, the passthrough function $t_{1i}(X) = X$ is thus used.

We thus obtain a vector of transformed deep features for each layer, as in Figure 2, of dimension $[O_c, 1]$. This is reduced to a low-dimensional vector of $c$ PCA coefficients in the next step of Fig. 2 for each of $N_L$ layers.

## 4.2 Transformed Feature Dimension Reduction

In general for an $N$ layer network we choose $N_L$ layers for transformation and dimensionality reduction, yielding $N_L$ vectors, each of dimension $[O_{c,i}, 1]$, where $i \in L$, and $L$ is the set of chosen layers (of cardinality $N_L$). A Principal Component Analysis is applied to each vector in the resultant set of $N_L$ vectors (Shlens, 2014). The PCA transform is fit to the set of such vectors for each layer obtained from the *clean* training data subset available to the defender (See assumptions made in Section 2.3). We have chosen to use the PCA for dimension reduction due to its computational efficiency, theoretical guarantees and interpretability and prior art such as (Cantareira et al., 2021) and (Zheng et al., 2022a) showing that relatively simple analyses such as channel Lipschitzness or deep feature statistics can provide substantial backdoor attack-relevant information. A brief discussion of this design choice is provided in Section 7.4.

The PCA projects each vector to an orthobasis, producing a vector of PCA coefficients ordered by the energy of their respective orthobasis components. We choose a few of these coefficients for outlier detection, fitting a distribution to the PCA coefficient values over the clean training data subset for detecting outlier (backdoor) behavior in deployment. Figs. 3c and 3d show the effectiveness of various chosen coefficients (first five, or five highest energy; and last ten coefficients, or ten lowest energy) for outlier detection on a test Trojan attack (the BPP attack (Wang et al., 2022b)) on CIFAR100 Preact ResNet-18. Here, the transform used was $t_{1i}(X) = Var(X)$ (as in Equation 5), where X is a 2-D feature map output from a convolutional kernel. The final (lowest-energy) PCA coefficients are seen to be the most effective, and the last PCA coefficient is chosen for our detection flow when using the variance transform.

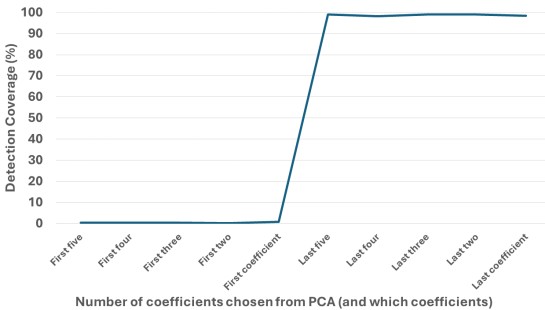

(a) Backdoor attack detection coverage when varying the number of PCA coefficients used, ranging from first five (five highest energy) to last five (five lowest energy).

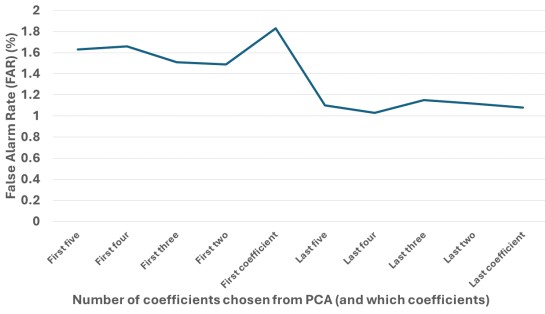

(b) Backdoor attack FAR when varying the number of PCA coefficients used, ranging from first five (five highest energy) to last five (five lowest energy).

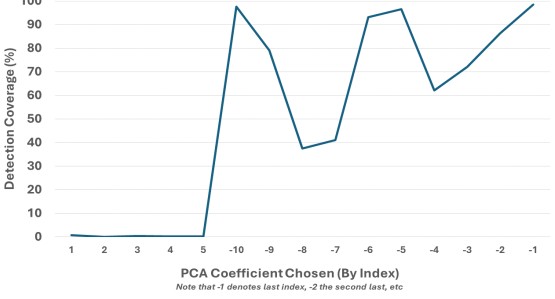

(c) Backdoor attack detection coverage when varying which single PCA coefficient is used, ranging from first five (five highest energy) to last ten (ten lowest energy).

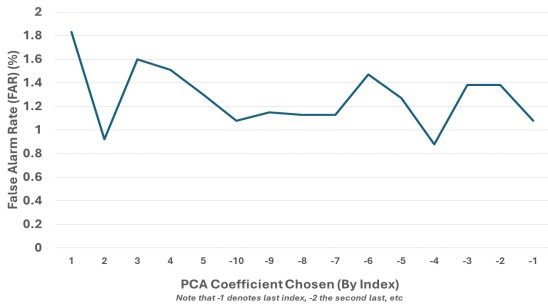

(d) Backdoor attack FAR when varying the single PCA coefficient used, from first five (five highest energy) to last ten (ten lowest energy).

Figure 3: Backdoor attack detection coverage and false alarm rate for varying the number and position (energy) of PCA coefficients, for Preactresnet-18 on CIFAR100, for the BPP attack (Wang et al., 2022b). The final (lowest energy) PCA coefficients give best detection coverage, while false alarm rate remains consistent for constant outlier detector hyperparameters. Using more coefficients does not appreciably increase detection coverage.

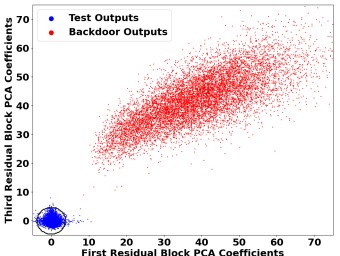 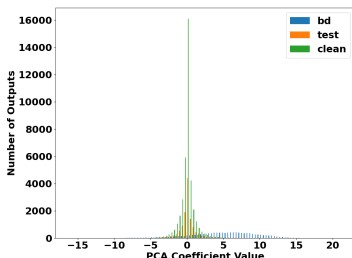 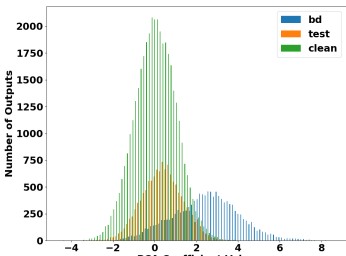

(a) Example of Gaussian MCD outlier detection on a clean and backdoored test dataset for the TrojanNN attack (Liu et al., 2018b) on Preactresnet-18 (Tiny-ImageNet (Le & Yang, 2015)).

(b) Example of last PCA coefficient of the transformed deep feature distribution shift between clean and backdoored test dataset (training dataset histogram present for comparison) for the Input-Aware Backdoor on Preactresnet-18 (GTSRB (Stallkamp et al., 2011)).

(c) Example of the last PCA coefficient of the transformed final layer output distribution shift between clean and backdoored test dataset (training dataset histogram present for comparison) for the Input-Aware Backdoor on Preactresnet-18 (GTSRB (Stallkamp et al., 2011)).

Figure 4: Examples of outlier detection and PCA coefficient distribution shift. Fig. 4b shows the importance of the deep features, where the narrower distribution and greater shift caused by an attack enabling high-coverage detection. The 'bd' in Fig. 4b and 4c indicates the backdoored test dataset distribution. The 'test' in those figures indicates the PCA coefficient distribution for the clean test dataset. The 'clean' indicates the clean training data distribution (used to train the elliptic envelope MCD). This way, we make use of the distribution of clean training data to perform outlier detection on backdoored inputs with no assumptions as to the backdoor trigger or injection mechanism

Multiple PCA coefficients can also be used for detection, as shown in Figs. 3a and 3b (using the first and last five coefficients). We see that taking 1-5 of the last (lowest-energy) PCA coefficients produces a minor increase in detection coverage and false alarm rate for the test attack, at the cost of linear scaling in computation requirements (as the outlier detector distribution dimension increases). The number of PCA coefficients used and the choice of coefficients thus remains a tunable parameter dependent on the transform $t_{1i}$.

Thus, using the PCA for dimensionality reduction, we have a second transform $t_{2i}$ applied to the $i$th layer deep features ($i \in L$), such that $t_{2i}(X) = (T_i.t_{1i}(X))_c$, where $c$ is the set of chosen PCA coefficients (for $t_{1i}(X)$ in Equation 5, the last coefficient) and $T_i$ is the PCA transform matrix at the $i$th layer. This thus yields $c$ PCA coefficients from a high-dimensional layer for outlier (backdoor) detection, giving a $cN_L$-dimensional vector $\theta$ for outlier detection, so that:

$$\theta = [t_{2i}(t_{1i}(f_i \circ f_{i-1} \circ \cdots \circ f_0(x^b)))]_{i \in L} \tag{6}$$

is the input to $Outlier(.)$ in the detector function of Equation 3.

## 4.3 Outlier Detection for Backdoor Attack Detection

To detect an attack we check if the vector obtained for a particular sample is an outlier with respect to the true (i.e., clean) distribution of $\theta$ as observed in the training set. This of course would only work if an attack appreciably alters the distribution of the PCA coefficients used in the construction of the vector $\theta$, which we do confirm empirically to hold true (Figures 4b and 4c). Fig. 4b shows the importance of the use of transformed deep features for TESDA, since its backdoored PCA coefficients are highly divergent from the clean distribution - more so than the case for the network outputs' PCA coefficient of Fig. 4c.

We then approximate the distribution of the clean $\theta$ across the training set as a multidimensional Gaussian $\mathcal{N}(\mu, \Sigma)$ (thus $\mathcal{D}^C$ in Section 3.2 is a multidimensional Gaussian), allowing us to leverage the statistical tools available for robust Gaussian parameter estimation and model fitting. Specifically, we use the minimum covariance determinant (MCD) method (Rousseeuw, 1984; 1985; Rousseeuw & Driessen, 1999) which given a training set with $n$ samples each of dimension $cN_L$ of which at most $m$ are outliers, finds estimates $\hat{\mu}$ and $\hat{\Sigma}$ for the true mean and covariance. To do so, the MCD finds $v = \frac{n+cN_L+1}{2}$ samples from the training set whose

covariance matrix has the least determinant and fits a Gaussian to them, implicitly making the assumption that $m \lesssim n - v = \frac{n - cN_L - 1}{2}$. Subsequently, any sample that is outside the Elliptic Envelope described by $\hat{\mu}$, $\hat{\Sigma}$, i.e., has a Mahalanobis distance $d = \sqrt{(\theta - \hat{\mu})^T \Sigma^{-1}(\theta - \hat{\mu})} \geq \Delta$, where $\Delta$ is a pre-specified threshold, is classified as an outlier. It should be noted that for Gaussian models, the MCD method has been shown to be asymptotically consistent (Butler et al., 1993), i.e., as $n \to \infty$, $\hat{\mu} \to \mu$ and $\hat{\Sigma} \to \Sigma$. An example of the MCD based elliptic envelope fit to noisy data processed through a DNN is shown in Figure 4a. Thus, in Equation 3, the function $Outlier(.)$ is:

$$Outlier(.) = MCD(\theta) = \begin{cases} 1 \text{ if } d \geq \Delta \\ 0 \text{ if } d < \Delta \end{cases} \tag{7}$$

where 1 denotes an outlier and 0 denotes an inlier.

### 4.3.1 Hyperparameter Tuning to Match Target False Positive Rate (FPR)

Due to its simplicity and interpretability, the MCD method lends itself to principled hyperparameter tuning which allows us to choose the offset $\Delta$ such that given an approximate upper bound on the degree of contamination of the training data, $\epsilon \in (0, 0.5)$, we can stay under either a target false negative (FNR) or false positive rate (FPR).

Let $\hat{\mu} \in \mathcal{R}^{cN_L}$ ($cN_L$, as before, is the dimension of the concatenation of $N_L$ $c$-dimensional PCA coefficient vectors) and positive definite $\hat{\Sigma} \in \mathcal{R}^{cN_L \times cN_L}$ be the robust sample mean and covariance estimates provided by the MCD method over the reduced-dimension transformed DNN training dataset deep features. Further assume $n > v \gg cN_L$ and $\hat{\mu} \approx \mu$, $\hat{\Sigma} \approx \Sigma$. For a sample $\theta$ to be classified as an outlier (i.e., an attack) we require that $d \geq \Delta$, or equivalently $d^2 > \Delta^2$, where $d^2 = (\theta - \hat{\mu})^T \Sigma^{-1}(\theta - \hat{\mu})$ is the squared Mahalanobis distance of $\theta$ and $\Delta$ is a constant computed over and fit to the entire training data. Specifically, $\Delta$ depends on the contamination parameter $\epsilon$ and is fit on the training data such that the number of outliers $m \approx \epsilon n$.

Given that $\epsilon$ acts as an upper bound on the contamination of the training set (the proportion of the training dataset that are outliers), we want to pick $\Delta^2$ such that $P[d^2 \geq \Delta^2] \leq \epsilon$. At the same time, by the multivariate Chebyshev inequality (Stellato et al., 2017) we have $P[d^2 \geq t^2] \leq \frac{cN_L(n^2 - 4 + 2nt^2)}{n^2 t^2}$. Equating the two, we get $\Delta = \sqrt{\frac{cN_L(n^2 - 4)}{\epsilon n^2 - 2ncN_L}}$. Provided this expression for $\Delta$ we have a principled way of choosing the contamination parameter $\epsilon$ so as to stay under a specified false negative ($\tau_F$) or false positive ($\tau_P$) rate, and fixing $\Delta$ from this $\epsilon$.

**False negative rate:** The FNR is by definition the complement of the true positive rate, which on the training set must be $\epsilon$. Therefore to minimize FNR, we set $\epsilon = 1 - \tau_F$ and then fix $\Delta \leq \sqrt{\frac{cN_L(n^2 - 4)}{\epsilon n^2 - 2ncN_L}} = \sqrt{\frac{cN_L(n^2 - 4)}{(1 - \tau_F)n^2 - 2ncN_L}}$.

**False positive rate:** To minimize false positives we simply require that for clean samples $P(d^2 \geq \Delta^2) \leq \tau_P$. We therefore set $\epsilon = \tau_P$ and fix $\Delta \geq \sqrt{\frac{cN_L(n^2 - 4)}{\epsilon n^2 - 2ncN_L}} = \sqrt{\frac{cN_L(n^2 - 4)}{\tau_P n^2 - 2ncN_L}}$.

The details regarding the derivation of the expression for $\Delta$ are given in Section 4.3.3.

### 4.3.2 Deriving Tighter Bounds for Outlier Detection

The squared Mahalanobis distance of a sample $\theta$ from the true distribution of mean $\mu$ and covariance $\Sigma$ is:

$$d^2 = (\theta - \mu)^T \Sigma^{-1}(\theta - \mu) \tag{8}$$

Since $\Sigma$ is symmetric positive definite, we can rewrite Equation 8 as:

$$d^2 = (\Sigma^{-\frac{1}{2}}(\theta - \mu))^T (\Sigma^{-\frac{1}{2}}(\theta - \mu)) \tag{9}$$

Setting $Y = \Sigma^{-\frac{1}{2}}(\theta - \mu)$ in Equation 9, $d^2 = Y^T Y = ||Y||_2^2$ where the elements of the vector $Y$, $Y_i \sim \mathcal{N}(0, 1)$ are i.i.d $\forall i \in [1, 2, ..., cN_L]$. This makes the distribution for $d^2$ the same as the sum of squares of $cN_L$ independent standard normal distributions, which is the definition of the $\chi^2_{cN_L}$ distribution.

Leveraging the fact that the distribution of $d^2$ for clean $\theta$ is given by the chi-squared distribution with $cN_L$ degrees of freedom, i.e., $\chi^2_{cN_L}$, it is possible to have bounds tighter than those using Chebyshev's inequality, with the caveats that i) the expression for $\Delta$ would have to be split into multiple expressions depending on its range, or ii) the dependence of $\Delta$ on $\epsilon$ might no longer be expressible as closed form expressions of elementary functions.

For example, noting that the distribution $\chi^2_{cN_L}$ is sub-exponential with parameters $(2cN_L, 4)$ (Ghosh, 2021), one can write the corresponding sub-exponential tail bound (Wainwright, 2019), resulting in the following expressions:

$$\Delta = \begin{cases} \sqrt{4cN_L\sqrt{ln(\frac{1}{\epsilon^2})} + cN_L} & \text{if } \sqrt{cN_L} \leq \Delta \leq \sqrt{c^2N_L^2 + cN_L} \\ \sqrt{8ln(\frac{1}{\epsilon}) + cN_L} & \text{if } \Delta > \sqrt{c^2N_L^2 + cN_L} \end{cases} \tag{10}$$

Alternatively, one could also write a Chernoff bound (Wainwright, 2019) that yields $\Delta = \sqrt{-cN_L W(\frac{-\epsilon^{\frac{2}{cN_L}}}{e})}$, $W(.)$ being the Lambert $W$ function (Bronstein et al., 2008) whose values may be calculated numerically. Given an expression for $\Delta$, setting $\epsilon = 1-\tau_F$ (or $\epsilon = \tau_P$) results in the value of $\Delta$ that matches a target FNR (or FPR) as described in Section 4.3.1. Details of the Chernoff bound derivation can be found in Appendix A.1.

### 4.3.3 Deriving a General Expression for $\Delta$ Dependent on $\epsilon$

Given $\hat{\mu}$, $\hat{\Sigma}$ as our sample mean and covariance estimates, for a sample $\theta_i \in \mathcal{R}^{cN_L}$ to be classified as an outlier we require that its squared Mahalanobis distance $d_i^2 \geq \Delta^2$. Here, $d_i^2 = (\theta_i - \hat{\mu})^T\hat{\Sigma}^{-1}(\theta_i - \hat{\mu})$ and $\Delta$ is a specified threshold computed over the training set such that the number of outliers present in it are consistent with the provided estimate. To that end, we know that no more than $m = \epsilon n$ samples of our training set can have their $d_i^2 \geq \Delta^2$. Assuming $d^2$ to be the random variable governing the squared Mahalanobis distances for over the training set, we have the inequality

$$P(d^2 \geq \delta^2) \leq \epsilon \tag{11}$$

Next, we note that the multivariate Chebyshev inequality in $cN_L$ dimensions, with estimated mean and variance over $v$ samples (Stellato et al., 2017), can be stated as:

$$P(d^2 \geq \delta^2) \leq \min(1, \frac{cN_L(v^2 - 1 + v^2\delta^2)}{v^2\delta^2}) \tag{12}$$

We shall only consider the non-trivial case where $\min(1, \frac{cL(v^2-1+v^2\delta^2)}{v^2\delta^2}) = \frac{cN_L(v^2-1+v^2\delta^2)}{v^2\delta^2}$, simplifying Equation 12 to:

$$P(d^2 \geq \delta^2) \leq \frac{cN_L(v^2 - 1 + v^2\delta^2)}{v^2\delta^2} \tag{13}$$

Substituting $\delta^2 = \Delta^2$ and recalling from Section 4.3 that when given $n$ as the total number of training data points, $v = \frac{n-cN_L-1}{2} \approx \frac{n}{2}$ since $n \gg cN_L$, we finally get:

$$P(d^2 \geq \Delta^2) \leq \frac{cN_L(n^2 - 4 + 2n\Delta^2)}{n^2\Delta^2} \tag{14}$$

Equating Equation 11 and Equation 14, we get $\epsilon = \frac{cN_L(n^2-4+2n\Delta^2)}{n^2\Delta^2} \implies \Delta = \sqrt{\frac{cN_L(n^2-4)}{\epsilon n^2 - 2ncN_L}}$, as in Section 4.3.1.

## 5 Experimental Setup

### 5.1 Experimental Test Cases

We have tested our backdoor attack detection framework on two major convolutional neural network (CNN) backbones, on three different datasets. The network backbones we have used are Preactresnet18 (He et al.,

| Attack Test Case | Trigger Injection Mechanism | Trigger Stealth (Visual Inspection) | Trigger Locality | Input Dependence |
|---|---|---|---|---|
| TrojanNN (Liu et al., 2018b) | Image Patch | Visually Apparent | Local | Independent (Fixed Trigger) |
| BPP (Wang et al., 2022b) | Color Channel Quantization | Stealthy | Nonlocal | Input-Dependent |
| Wanet (Nguyen & Tran, 2021) | Warping Transform | Stealthy | Nonlocal | Input-Dependent |
| SSBA (Li et al., 2021c) | Additive Noise | Stealthy | Nonlocal | Input-Dependent |
| Input-Aware Backdoor Nguyen & Tran (2020) | GAN-generated perturbation | Visually Apparent | Nonlocal | Input-Dependent |

Table 1: Test Case Attacks examined in Section 6. We have chosen attacks to examine a diversity of trigger transforms, local/nonlocal triggers and trigger stealth (whether the trigger is visible on visual inspection of input images). All triggers have been generated using the benchmark repository in BackdoorBench for reproducibility (Wu et al., 2022).

2015) and VGG-19BN (Simonyan & Zisserman, 2014). The datasets we have used are (1) The GTSRB dataset (Stallkamp et al., 2011), consisting of 43 different traffic signs (classes) across 39,209 32x32 training and 12,360 test images; (2) The CIFAR100 dataset (Krizhevsky, 2009), which consists of 100 different classes across 50,000 32x32 training and 10,000 test images; and (3) The Tiny-ImageNet dataset (Le & Yang, 2015), consisting of 200 classes of 64x64 images, 100,000 training and 10,000 test images. The CNN backbones chosen lightweight CNN models (e.g., PreactResNet-18 and VGG-19BN) that are commonly used in edge applications, paralleling those employed in the prior art (Gao et al., 2019; Zheng et al., 2022a;b; Liu et al., 2018b). These architectures allow us to examine the effects of characteristics such as convolutional feature extraction, residual/dense connection blocks, and batch normalization layers on TESDA and its capture of the shifts in deep feature distributions.

We have tested a wide range of attacks, selected to provide a range of trigger functions and trigger positions. As in Section 2.1, a *local* image trigger is one that is localized to a particular region of the input image (e.g. a single square patch). A *nonlocal* trigger does not have a restriction on its location or size. The attacks we have tested are part of the overview in Section 2.1 (Tabulated in Table 1):

- The TrojanNN attack (Liu et al., 2018b), providing a localized, fixed trigger superimposed on the input image that is readily apparent on visual inspection.

- The Input-aware Dynamic Backdoor (Nguyen & Tran, 2020), providing a nonlocal, input-dependent trigger superimposed on the input image. These triggers are generated using GANs, using multiple perturbations at different points in the input image to generate unique triggers for each image.

- The BPP attack (Wang et al., 2022b), providing a nonlocal, input-dependent transform-based trigger applied to the input image. This is an imperceptible trigger that perturbs color channel quantization coefficients to cause misclassification, affecting the entire image while minimizing magnitude of disturbance.

- The Single-Sample Based Attack (SSBA) (Li et al., 2021c), using sample-specific additive noise encoded into images (nonlocal, input-dependent trigger). This trigger is likewise unique to the image (the sample) and imperceptible on visual inspection.

- The WaNet (Nguyen & Tran, 2021) attack, using a nonlocal input-dependent warping transform to generate a trigger for the input image. WaNet is imperceptible, applies warping transforms across the image (nonlocal disturbance) and acts to minimize perturbation magnitude.

These are applied to multiple baselines for comparison to TESDA, along with the base case of no defense or detection mechanism in place (to test attack effectiveness). The chosen attack test cases allow us to test GAN-based triggers, local/nonlocal triggers, different trigger transforms (warping, additive noise, color channel quantization) and trigger perturbation stealth and its effectiveness on detection.

## 5.2 Baselines

Our experimental baselines are likewise selected to provide a mix of offline defense mechanisms and detection systems, and online detection methods for comparison. A comparison of these methods to other state of the art work can be found in Section 2.2. The baselines are:

- Spectral signature based detection (Tran et al., 2018) uses the L2 norm of the top singular vector of each class, comparing the clean (training) class spectrum with the poisoned training spectrum.

- STRIP (Gao et al., 2019) is an on-line detection mechanism that uses the entropy distribution of perturbed final layer outputs of the DNN, and is present as an on-line baseline. We have calculated entropy thresholds for STRIP using a two-sided Student's $t$-test rather than simply using a fixed bound based on training data values to enhance its performance.

- Minibatch norm statistics-based pruning (MBNS) (Zheng et al., 2022b) uses the distribution of mini-batch norm statistics to prune neurons most likely to contribute to misprediction from a backdoor.

- CLP prunes neurons based on channel lipschitzness bounds (Zheng et al., 2022a), since backdoored neurons cause uncharacteristic deep feature outputs which violate said bounds.

STRIP is used as an online baseline, Spectral is used to evaluate a method using output singular values (but not deep features) for detection, MBNS and CLP are recent work that use deep feature-based bounds (either minibatch norm statistics or lipschitzness) for comparison. All baselines except STRIP were implemented using the BackdoorBench (Wu et al., 2022) benchmarking repository. STRIP was implemented using code provided by the authors of the Input-Aware backdoor paper in (Nguyen & Tran, 2020).

### 5.3 Metrics and Ablations

For each test case, we record the detection coverage (true positive rate, TPR) for TESDA on a backdoored test dataset, and the false positive rate (FPR) for TESDA on a clean test dataset. For the STRIP baseline, we do the same. For Spectral, MBNS and CLP, we record the success rate (percentage of attacks stopped) after the defenses are used (i.e. defense effectiveness), and the network accuracy loss on clean test data relative to the network without any defense mechanism, after the defense is used (the cost of the defense, analogous to FPR of a detector that rejects inputs it flags as backdoored).

By default, TESDA uses the outputs from each residual block in Preactresnet-18 and the outputs from each batchnorm layer in VGG-19BN for transformation and dimension reduction. We have chosen these layers to balance deep feature information (the outputs of functional blocks or post-normalization) and overhead of TESDA (minimal number of layers chosen). It also uses the outputs of each dense layer, if they are present, and the network output. These are subject to PCA directly (as detailed in Section 2). By default, TESDA uses feature map variance as its convolutional layer transform (Equation 5).

The ablations we have performed are as follows:

- We study the use of feature map entropy as the TESDA transform instead of feature map variance and use a different set of PCA coefficients for detection in Section 6.3.

- We change TESDA to use the outputs of every layer in the CNN (ablation in Section 6.1). This provides greater detection coverage and catches attacks such as the SSBA who cause large shifts in deep features that may not be apparent when using just residual block or batchnorm outputs.

- We evaluate the size of the training data needed for fitting the TESDA detector distribution in Section 6.2.

- We evaluate the end to end latency of TESDA compared to STRIP applied to VGG19BN and Preactresnet-18 in Section 6.4.

## 6 Experimental Results and Ablations

Tables 2-4 illustrate the performance of TESDA in comparison to the baselines (see Section 5.2) on the networks, attacks and datasets under consideration from Section 5.1. This section will compare the effects of different attacks, datasets and network backbones on TESDA's performance, in comparison to the state of the art. We have used a contamination parameter $\epsilon = 0.01$ for the MCD outlier detector, and the last PCA

coefficient (lowest energy). TESDA in this 'standard' configuration uses the feature map variance (Eqn. 5) as its transform for convolutional layers, a passthrough function for dense layers, and is placed after each residual block, after the first convolutional layer and after the output layer (for Preactresnet-18) or after each batchnorm or dense layer (for VGG-19BN).

*Dataset effects:* Table 2 shows the TPR and FPR rate for TESDA on GTSRB, Table 3 shows those metrics for CIFAR-100 and Table 4 shows the data for the Tiny Imagenet dataset. We see that detection coverage is near-total for the GTSRB dataset, but FPR is also highest - in some cases breaching the theoretical FPR bound set in Section 4.3.1 due to the unbalanced nature of the GTSRB dataset. Tiny-Imagenet detection in Table 4 shows consistently high detection and low FPR, better than for GTSRB, with the greater number of dataset samples and larger input size allowing the transform-based dimension reduction and distribution fit to more accurately model clean network behavior (and forcing attacks to make larger changes to network behavior to force misprediction).

CIFAR-100 shows a lower TPR than Tiny Imagenet or GTSRB due to its lower resolution, higher class granularity and increased noise, which collectively yield more variable deep feature representations. The lower resolution (32x32) compared to the other two datasets renders deep features and hence extracted PCA representations more sensitive to noise, which complicates the process of detecting 'outlier' attack-related behavior. This is compounded by intraclass granularity, with 100 classes. This fine-grained categorization results in less robust, well-separated feature distributions. As a consequence, the transformation process via PCA – which critically depends on a stable variance structure – faces greater difficulty in establishing a clear detection threshold. In contrast, datasets like Tiny ImageNet and GTSRB provide more distinctive features per image, assisting the outlier detection process. These factors, together with smaller size, higher noise and the higher relative impact of a backdoor trigger on a smaller image render establishment of a stable decision boundary for attack (outlier) detection more difficult than for Tiny Imagenet or GTSRB.

*Effects of Network Type:* We see that in Tables 2-4, detection coverage for VGG-19BN is higher than for Preactresnet-18, due to VGG-19BN's deep structure and lack of residual connections making deviations from the nominal distribution of reduced-dimension transformed deep features more easily apparent. The higher FPR (breaching theoretical target FPR) for VGG-19BN on GTSRB also shows that the VGG backbone is less resilient to unbalanced datasets and small datasets than Preactresnet due to its deep structure and the vanishing gradient problem.

*Effects of Attack Type:* As expected, the fixed localized triggers for TrojanNN are the most easily detected by all baselines and by TESDA, with TESDA achieving near total detection coverage for all cases except Preactresnet-18 on CIFAR-100. The transform-based BPP attack also sees very high coverage and low FPR for TESDA, beating out the other baselines, due to the BPP transform causing high-frequency (low-energy) changes in the image that are more easily apparent at the final PCA coefficients. The input-aware backdoor is similar in coverage but slightly less, being a trigger function that applies a mask to each image rather than applying a transform like the BPP.

## 6.1 TESDA Layer Ablations

However, some attacks cause shifts in the reduced-dimension transformed deep feature distribution at layers that are not covered by the 'default' TESDA configuration used in the earlier section. In this section we examine the use of TESDA for all layers of the DNN in question, using $\epsilon = 0.03$. This is done for the SSBA and the WaNet attack, both of which are seen to evade TESDA at the earlier configuration. Table 5 shows the detection results for the CIFAR-100 dataset compared to our baselines, while Table 6 shows corresponding data for the Tiny Imagenet dataset.

We see that the use of all layers achieves significant improvement over the state of the art and over the 'default' TESDA configuration for Tiny Imagenet. However, attacks such as WaNet and SSBA that rely on warping transforms or encoding-based triggers are able to leverage the properties of CIFAR-100 and evade TESDA - forcing TPR to as low as 63% for Preactresnet-18 on CIFAR100, and likewise evading the baselines. The use of all layers also results in breaching the theoretical FPR bounds for CIFAR-100, likely due to nongaussian behavior for some of the reduced-dimension transformed deep feature distributions. This does not occur for Tiny Imagenet.

| | | TrojanNN | | BPP | | Input-Aware | |
|---|---|---|---|---|---|---|---|
| | | Preactresnet-18 | VGG-19BN | Preactresnet-18 | VGG-19BN | Preactresnet-18 | VGG-19BN |
| No Defense | Attack Success Rate (%) | 100 | 100 | 99.9 | 63.53 | 95.92 | 85.03 |
| | Clean Accuracy (%) | 98.57 | 97.77 | 97.43 | 97.82 | 98.76 | 96.32 |
| TESDA | TPR (%) | **100** | **100** | 9.87 | 8.37 | 0.8 | 9 |
| | FAR (%) | 2.29 | 8.9 | 2.7 | 4.82 | 3 | 5 |
| STRIP | TPR (%) | 9.9 | 0.3 | 69.89 | 20.67 | 24.41 | 30.8 |
| | FAR (%) | 1.83 | 2.51 | 2.56 | 1.83 | 2 | 2.7 |
| Spectral | Defense Success Rate (%) | 0 | 0 | 8.86 | 14.71 | 22.94 | 36.28 |
| | Loss in Accuracy (%) | 0.84 | 0.87 | 0.25 | 1.2 | 0.63 | .06 |
| CLP | Defense Success Rate (%) | 4.8 | 0.54 | 16.66 | 96.98 | 3.99 | 53.87 |
| | Loss in Accuracy (%) | .4 | .11 | -0.03 | .1 | .08 | -0.55 |
| MBNS | Defense Success Rate (%) | 0 | 1.47 | **100** | **100** | 99.94 | 98.97 |
| | Loss in Accuracy (%) | **0.06** | **-0.01** | **-0.01** | **0.02** | **-0.08** | **-0.67** |

Table 2: Attack detection results for GTSRB dataset. The highest detection coverage or defense success rate (TPR) is bolded, and the lowest loss in accuracy or FPR is bolded, for each column. The second highest detection coverage or defense success rate (TPR) is underlined, and the second lowest loss in accuracy or FPR is underlined, for each column. The MBNS defense - and other offline defense baselines achieve the lowest FPR but fail to detect all attacks consistently even on GTSRB. TESDA shows the highest or second highest TPR, while in almost all cases maintaining a low FPR.

| | | TrojanNN | | BPP | | Input-Aware | |
|---|---|---|---|---|---|---|---|
| | | Preactresnet-18 | VGG-19BN | Preactresnet-18 | VGG-19BN | Preactresnet-18 | VGG-19BN |
| No Defense | Attack Success Rate (%) | 100 | 99.98 | 98.87 | 98.28 | 98.63 | 85.52 |
| | Clean Accuracy (%) | 69.82 | 64.89 | 64 | 60.13 | 65.24 | 59.45 |
| TESDA | TPR (%) | 6.81 | **99.9** | 8.5 | 1.32 | **99.6** | 61.7 |
| | FAR (%) | .14 | .75 | | .2 | .56 | **1** |
| STRIP | TPR (%) | **99.9** | 6.3 | 56.58 | 39.81 | 74.15 | 80.9 |
| | FAR (%) | 5 | 2.47 | 3.86 | 2.96 | 3.94 | 2.01 |
| Spectral | Defense Success Rate (%) | 0.07 | 0.01 | 0.73 | 8.55 | 6.18 | 10.59 |
| | Loss in Accuracy (%) | 4.36 | 5.32 | **-0.89** | 2.27 | -1.74 | 2.18 |
| CLP | Defense Success Rate (%) | 0.19 | 59.82 | **99.65** | **99.91** | 1.94 | 9.6 |
| | Loss in Accuracy (%) | 1.6 | 10.57 | 4.18 | 4.53 | 1.91 | 1.73 |
| MBNS | Defense Success Rate (%) | 0 | 22.48 | 96.5 | 6.2 | 9.6 | **100** |
| | Loss in Accuracy (%) | **0.44** | **1.16** | 2.37 | **0.96** | 1.44 | .61 |

Table 3: Attack detection results for the CIFAR100 dataset. The highest detection coverage or defense success rate (TPR) is bolded, and the lowest loss in accuracy or FPR is bolded, for each column. The second highest detection coverage or defense success rate (TPR) is underlined, and the second lowest loss in accuracy or FPR is underlined, for each column. TESDA maintains a high FPR and second lowest or lowest FPR/accuracy loss among the baselines. The offline pruning based MBNS method shows total effectiveness for some attacks but fails for others - much less consistent performance.

| | | TrojanNN | | BPP | | Input-Aware | |
|---|---|---|---|---|---|---|---|
| | | Preactresnet-18 | VGG-19BN | Preactresnet-18 | VGG-19BN | Preactresnet-18 | VGG-19BN |
| No Defense | Attack Success Rate (%) | 99.98 | 99.97 | 100 | 99.96 | 98.04 | 99.84 |
| | Clean Accuracy (%) | 55.89 | 52 | 58.14 | 55.36 | 57.78 | 53.2 |
| TESDA | TPR (%) | **100** | 9.36 | **99.98** | **100** | 97.2 | **100** |
| | FAR (%) | 0.85 | 0.73 | | 0.58 | 0.96 | 0.9 |
| STRIP | TPR (%) | 7.96 | **99.38** | 57.49 | 98.43 | 9.15 | 35.8 |
| | FAR (%) | 5.66 | 2.31 | 2.5 | 1.8 | 2.13 | 2.26 |
| Spectral | Defense Success Rate (%) | 0 | 0.06 | 0.46 | 1.07 | 0.41 | 1.39 |
| | Loss in Accuracy (%) | 3.58 | 6.35 | 6.27 | 9.66 | 5.95 | 7.38 |
| CLP | Defense Success Rate (%) | 91.61 | 62.76 | 9.72 | 9.83 | 0.42 | 99.66 |
| | Loss in Accuracy (%) | .03 | .03 | 1.11 | .14 | **0.03** | -0.2 |
| MBNS | Defense Success Rate (%) | 0.02 | 0.03 | 0.02 | 0.05 | 0.42 | 9.94 |
| | Loss in Accuracy (%) | **0** | **0** | 0.48 | 0.13 | .06 | **-0.4** |

Table 4: Attack detection results for the Tiny Imagenet dataset. The highest detection coverage or defense success rate (TPR) is bolded, and the lowest loss in accuracy or FPR is bolded, for each column. The second highest detection coverage or defense success rate (TPR) is underlined, and the second lowest loss in accuracy or FPR is underlined, for each column. TESDA consistently delivers near-total TPR and a low FPR within its theoretical bounds ($\epsilon = 0.01$). The offline defenses like MBNS and CLP deliver high effectiveness only for one or two cases, and while they maintain a low accuracy loss, this is at the cost of collapse in defense effectiveness. The highest TPR/defense effectiveness and lowest FPR/accuracy loss in each category are bolded, and the second highest are underlined. Methods such as MBNS that prune and then re-train the network may also increase network nominal accuracy, as we see here, although their effect on backdoor detection is variable.

| | | WaNet | | SSBA | |
|---|---|---|---|---|---|
| | | Preactresnet-18 | VGG-19BN | Preactresnet-18 | VGG-19BN |
| No Defense | Attack Success Rate (%) | 97.73 | 96.22 | 96.51 | 95.74 |
| | Clean Accuracy (%) | 64.05 | 57.91 | 69.26 | 64.38 |
| TESDA | TPR (%) | 5.6 | 64.31 | 8.41 | 69.73 |
| | FAR (%) | 5.63 | 4.29 | 5.63 | 6.74 |
| TESDA (all layers) | TPR (%) | 62.94 | 77.4 | 78.34 | 90.25 |
| | FPR (%) | 3.27 | 5.15 | 4.16 | 5.13 |
| STRIP | TPR (%) | 18.86 | 12.52 | 46.99 | 0.59 |
| | FAR (%) | 2.79 | 3.63 | 1.08 | 2.08 |
| Spectral | Defense Success Rate (%) | 7.56 | 23.08 | 3.88 | 7.05 |
| | Loss in Accuracy (%) | -2.38 | 0.92 | 2.29 | 6.23 |
| CLP | Defense Success Rate (%) | 16.51 | 97.66 | 2.48 | 15.76 |
| | Loss in Accuracy (%) | 38.15 | -0.56 | 3.87 | 10.57 |
| MBNS | Defense Success Rate (%) | 3.7 | 9.9 | 3.17 | 5.39 |
| | Loss in Accuracy (%) | 1.25 | 11.33 | 0.26 | 4.28 |

Table 5: Attack detection coverage for WaNet and SSBA on CIFAR100 when using all layers for deep feature reduced-dimension transformation (rather than the subset used in Tables 2-4). TESDA using all layers achieves higher TPR than the baselines, at the cost of a slightly higher FPR than the baselines (whose defense effectiveness or TPR collapses).

| | | WaNet | | SSBA | |
|---|---|---|---|---|---|
| | | Preactresnet-18 | VGG-19BN | Preactresnet-18 | VGG-19BN |
| No Defense | Attack Success Rate (%) | 99.48 | 99.98 | 97.69 | 98.09 |
| | Clean Accuracy (%) | 56.6 | 54.11 | 55.31 | 51.2 |
| TESDA | TPR (%) | 56.4 | 68.09 | 7.46 | 23.66 |
| | FAR (%) | 2.66 | 3.31 | 3.12 | 3.47 |
| TESDA (all layers) | TPR (%) | 98.58 | 100 | 98.85 | 98.83 |
| | FPR (%) | 2.35 | 3.17 | 2.43 | 2.93 |
| STRIP | TPR (%) | 4.99 | 68.4 | 10.71 | 99.74 |
| | FAR (%) | 2.51 | 2.26 | 2.6 | 3.17 |
| Spectral | Defense Success Rate (%) | 0.54 | 2.4 | 2.04 | 1.95 |
| | Loss in Accuracy (%) | 4.21 | 15.31 | 3.7 | 5.52 |
| CLP | Defense Success Rate (%) | 1.5 | 0.01 | 2.35 | 1.67 |
| | Loss in Accuracy (%) | 0.39 | 0.59 | 0.14 | -0.29 |
| MBNS | Defense Success Rate (%) | 0.54 | 0.02 | 2.2 | 1.65 |
| | Loss in Accuracy (%) | 0.2 | 0.12 | 0.61 | 1.43 |

Table 6: Attack detection coverage for WaNet and SSBA on Tiny ImageNet when using all layers for deep feature reduced-dimension transformation (rather than the subset used in Tables 2-4). TESDA using all layers achieves higher TPR than the baselines, at the cost of a slightly higher FPR than the baselines (whose defense effectiveness or TPR collapses).

| Attack | | Input-Aware | | BPP | | TrojanNN | |
|---|---|---|---|---|---|---|---|
| Network | | Preactresnet18 | VGG-19BN | Preactresnet18 | VGG-19BN | Preactresnet18 | VGG-19BN |
| 50% of dataset used for fit | Mean TPR ± Standard Deviation (%) | 91.24 ± 0.81 | 99.89 ± 0.3 | 99.88 ± 0.06 | 99.86 ± 0.16 | 100 ± 0 | 100 ± 0 |
| | Mean FAR ± Standard Deviation (%) | 3.24 ± 0.24 | 6.14 ± 0.35 | 2.9 ± 0.17 | 5.52 ± 0.39 | 2.5 ± 0.19 | 9 ± 0.72 |
| 25% of dataset used for fit | Mean TPR ± Standard Deviation (%) | 91.57 ± 1.9 | 100 ± 0 | 99.92 ± 0.05 | 99.12 ± 0.46 | 100 ± 0 | 100 ± 0 |
| | Mean FAR ± Standard Deviation (%) | 3.6 ± 0.27 | 7.53 ± 0.56 | 3.48 ± 0.37 | 6.95 ± 0.41 | 3.08 ± 0.46 | 11.13 ± 0.61 |
| 10% of dataset used for fit | Mean TPR ± Standard Deviation (%) | 90.67 ± 2.9 | 100 ± 0 | 99.94 ± 0.07 | 99.64 ± 0.25 | 99.99 ± 0 | 100 ± 0 |
| | Mean FAR ± Standard Deviation (%) | 5.39 ± 0.5 | 12.93 ± 0.87 | 5.7 ± 0.73 | 12.09 ± 1.2 | 4.91 ± 0.66 | 16.32 ± 0.96 |

Table 7: Detection coverage for TESDA (standard configuration, as in Tables 2-4) for varying proportions (randomly sampled) of the GTSRB dataset used for fitting the outlier detector distribution. The mean and standard deviation of TPR and FAR are recorded. The values for the full dataset can be seen in Table 2.

| Attack | | Input-Aware | | BPP | | TrojanNN | |
|---|---|---|---|---|---|---|---|
| Network | | Preactresnet-18 | VGG-19BN | Preactresnet-18 | VGG-19BN | Preactresnet-18 | VGG-19BN |
| 50% of dataset used for fit | Mean TPR ± Standard Deviation (%) | 99.66 ± 0.08 | 73.92 ± 8.1 | 98.17 ± 0.72 | 92.71 ± 0.58 | 69.92 ± 1.21 | 99.92 ± 0.03 |
| | Mean FAR ± Standard Deviation (%) | 1.72 ± 0.09 | 1.22 ± 0.13 | 1.18 ± 0.08 | 1.28 ± 0.09 | 3.77 ± 0.07 | 2.12 ± 0.18 |
| 25% of dataset used for fit | Mean TPR ± Standard Deviation (%) | 99.66 ± 0.15 | 85.38 ± 4.18 | 96.93 ± 1.75 | 92.95 ± 2.67 | 71.54 ± 5 | 99.95 ± 0.03 |
| | Mean FAR ± Standard Deviation (%) | 1.88 ± 0.17 | 1.45 ± 0.14 | 1.32 ± 0.18 | 1.52 ± 0.14 | 1.31 ± 0.08 | 2.91 ± 0.34 |
| 10% of dataset used for fit | Mean TPR ± Standard Deviation (%) | 99.73 ± 0.3 | 91.43 ± 3.14 | 97.26 ± 3.43 | 97.2 ± 1.26 | 75.16 ± 7.53 | 99.96 ± 0.02 |
| | Mean FAR ± Standard Deviation (%) | 2.7 ± 0.2 | 3.01 ± 0.61 | 2.24 ± 0.19 | 3 ± 0.6 | 2.18 ± 0.21 | 7.44 ± 1 |

Table 8: Detection coverage for TESDA (standard configuration, as in Tables 2-4) for varying proportions (randomly sampled) of the CIFAR100 dataset used for fitting the outlier detector distribution. The mean and standard deviation of TPR and FPR are recorded. The values for the full dataset can be seen in Table 3.

## 6.2 Dataset Fraction Ablations

Tables 7-9 show the results of using only a subset of the training dataset (for GTSRB, CIFAR-100 and Tiny Imagenet) to fit the MCD for the reduced-dimension transformed deep features. We have varied the proportion of the dataset used from the full training dataset (data seen in Tables 2-4) to 50%, 25% and 10% of the training dataset. This is randomly sampled and the experiment repeated ten times with the mean of TPR and FPR recorded. To compare with the baselines, use the results in Tables 2-4. The MCD here uses $\epsilon = 0.01$.

We see that the TPR and FPR remain stable with very low standard deviation for TESDA until 10% of the dataset is used. For CIFAR-100 we see the FPR rise above the rate for GTSRB and Tiny Imagenet, similar to Tables 2-4. The TPR falls slightly as we move from 50% of the training dataset to 10%, and the FPR rises sharply for GTSRB and CIFAR-100, likely due to a less accurate distribution fit. The FPR rises much less for Tiny Imagenet as the proportion of dataset falls, as the number of samples remains larger in absolute terms compared to CIFAR-100 and GTSRB.

| Attack | | Input-Aware | | BPP | | TrojanNN | |
|---|---|---|---|---|---|---|---|
| Network | | Preactresnet-18 | VGG-19BN | Preactresnet-18 | VGG-19BN | Preactresnet-18 | VGG-19BN |
| 50% of dataset used for fit | Mean TPR ± Standard Deviation (%) | 97.34 ± 0.15 | 100 ± 0 | 99.99 ± 0 | 100 ± 0 | 100 ± 0 | 99.42 ± 0.09 |
| | Mean FAR ± Standard Deviation (%) | 1.05 ± 0.04 | 1.04 ± 0.04 | 1.02 ± 0.05 | 0.58 ± 0.04 | 0.85 ± 0.03 | 0.77 ± 0.03 |
| 25% of dataset used for fit | Mean TPR ± Standard Deviation (%) | 97.97 ± 0.59 | 99.99 ± 0 | 99.98 ± 0.01 | 100 ± 0 | 100 ± 0 | 99.53 ± 0.13 |
| | Mean FAR ± Standard Deviation (%) | 1.11 ± 0.08 | 1.11 ± 0.11 | 1.07 ± 0.06 | 0.67 ± 0.08 | 0.87 ± 0.04 | 0.8 ± 0.06 |
| 10% of dataset used for fit | Mean TPR ± Standard Deviation (%) | 98.08 ± 0.72 | 99.99 ± 0.01 | 99.98 ± 0.02 | 99.99 ± 0 | 100 ± 0 | 99.56 ± 0.34 |
| | Mean FAR ± Standard Deviation (%) | 1.32 ± 0.14 | 1.4 ± 0.1 | 1.29 ± 0.1 | 0.69 ± 0.15 | 1.03 ± 0.11 | 1.14 ± 0.18 |

Table 9: Detection coverage for TESDA (standard configuration, as in Tables 2-4) for varying proportions (randomly sampled) of the Tiny Imagenet dataset used for fitting the outlier detector distribution. The mean and standard deviation of TPR and FPR are recorded. The values for the full dataset can be seen in Tables 4.

| | | TESDA | | TESDA-E | | TESDA-E2 | |
|---|---|---|---|---|---|---|---|
| | | TPR (%) | FAR (%) | TPR (%) | FAR (%) | TPR (%) | FAR (%) |
| TrojanNN | Preactresnet-18 | 66.81 | 1.14 | 94.43 | 2.18 | 94.83 | 1.26 |
| | VGG-19BN | 99.9 | 1.75 | 99.96 | 2.9 | 99.94 | 1.49 |
| Input-Aware Backdoor | Preactresnet-18 | 99.6 | 1.56 | 99.92 | 3.45 | 99.93 | 1.3 |
| | VGG19-BN | 61.7 | 1 | 67.31 | 2.26 | 56.72 | 1.2 |
| BPP | Preactresnet-18 | 98.5 | 1 | 99 | 2.81 | 99.24 | 1.31 |
| | VGG-19BN | 91.32 | 1.2 | 46.87 | 2.26 | 24.13 | 1.18 |

Table 10: Comparisons of different transforms used for TESDA on CIFAR-100. The 'standard' TESDA algorithm uses the feature map variance for convolutional layer outputs (Eqn. 5) and the final PCA coefficient. TESDA-E uses feature map entropy (Eqn. 4) and takes the first and final (highest and lowest energy) PCA coefficients from each layer, using one outlier detector. TESDA-E2 uses feature map entropy and the first and final PCA coefficients, but uses one MCD outlier detector for each PCA coefficient set - if *either* detector flags a positive, the input is deemed backdoored. As such, its TPR is higher, as is its FPR. For the baseline metrics as comparison, see Tables 3.

| | | TESDA | | TESDA-E | | TESDA-E2 | |
|---|---|---|---|---|---|---|---|
| | | TPR (%) | FAR (%) | TPR (%) | FAR (%) | TPR (%) | FAR (%) |
| TrojanNN | Preactresnet-18 | 100 | 0.85 | 100 | 2.69 | 100 | 1.15 |
| | VGG-19BN | 99.36 | 0.73 | 99.93 | 1.85 | 99.82 | 0.96 |
| Input-Aware Backdoor | Preactresnet-18 | 97.2 | 0.96 | 99.93 | 2.85 | 99.92 | 1.21 |
| | VGG19-BN | 100 | 0.9 | 100 | 2.71 | 100 | 1.21 |
| BPP | Preactresnet-18 | 99.98 | 1 | 100 | 1.96 | 100 | 0.954 |
| | VGG-19BN | 100 | 0.58 | 100 | 2.09 | 99.96 | 1.01 |

Table 11: Comparisons of different transforms used for TESDA on Tiny ImageNet. The 'standard' TESDA algorithm uses the feature map variance for convolutional layer outputs (Eqn. 5) and the final PCA coefficient. TESDA-E uses feature map entropy (Eqn. 4) and takes the first and final (highest and lowest energy) PCA coefficients from each layer, using one outlier detector. TESDA-E2 uses feature map entropy and the first and final PCA coefficients, but uses one MCD outlier detector for each PCA coefficient set - if *either* detector flags a positive, the input is deemed backdoored. As such, its TPR is higher, as is its FPR. For the baseline metrics as comparison, see Table 4.

## 6.3 Transform Ablations

Tables 10 (for CIFAR-100) and 11 (for Tiny Imagenet) show the TPR and FPR for the 'default' TESDA compared against two other transform configurations. The data for the baselines to compare against this can be found in Tables 3 and 4 and are omitted here for brevity. The MCD here uses $\epsilon = 0.01$.

The TESDA-E configuration in the tables uses feature map entropy (Equation 4) as the transform applied to each convolutional layer output, and takes the first and last PCA coefficients (instead of just the final PCA coefficient). Each layer thus generates two PCA coefficients, and a $2N_L$-length vector is used for the MCD outlier detector. The TESDA-E2 configuration likewises uses feature map entropy as a transform, but feeds the first and last PCA coefficients into separate MCD detectors. These two outlier detectors both independently flag a backdoored input. TESDA-E2 thus flags a backdoor if *either* of its MCD outlier detectors flags an outlier.

We see that the 'default' TESDA performs roughly similarly to TESDA-E and TESDA-E2 for Tiny Imagenet test cases, but the entropy-based transforms outperform TESDA (at the cost of more computation required for feature map entropy and a larger-dimension MCD fit) for CIFAR-100. The entropy transform is able to better capture image features for each deep feature map, and a PCA reduction of feature map entropy that takes first and final PCA coefficients captures more relevant information than the simple variance transform used in the 'default' TESDA.

## 6.4 Detector Overhead

Figure 5 illustrates the average and standard deviation of end to end latency in milliseconds for the CNN using TESDA compared to the CNN using the STRIP online detector baseline. Fig. 5a details the mean latency, and Fig. 5b shows the standard deviation. The mean and standard deviation are calculated over 500 inputs streamed through the DNN (batch size of 1). These figures include CNN latency, and indicate end to end (classification and backdoor detector) latency. The CNN was run on an NVIDIA Quadro RTX 5000 (2018 model) and with an Intel Xeon W-2123 CPU (2018 model) using 32GB of RAM.

TESDA outperforms STRIP despite using only off-the-shelf functions from Scikit-learn and therefore doing its computations off the GPU rather than onboard (losing the acceleration provided by PyTorch). Its latency ranges from half or much less than STRIP on average, and in all cases save one (VGG-19BN on CIFAR-100) its standard deviation of latency is less. It also shows roughly consistent latency as dataset size and input sizes

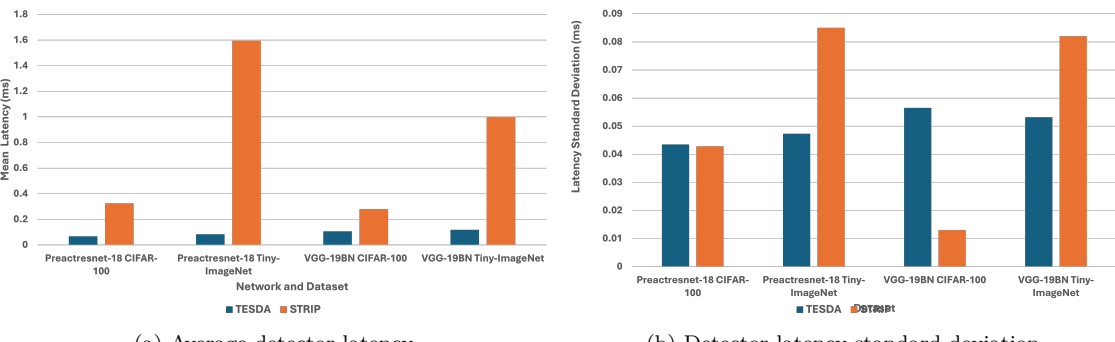

(a) Average detector latency.

(b) Detector latency standard deviation.

Figure 5: Detection latency averaged over 500 inputs, and detector latency standard deviation over those inputs, comparing STRIP to TESDA. TESDA shows consistently lower latency compared to STRIP, by a wide margin.

change from Tiny Imagenet, while STRIP sees sharp increases in latency as input sizes rise in Tiny Imagenet due to its entropy calculations and image superposition transforms. We note that the standard deviations of latency for TESDA and STRIP are both very low (order of hundredths of milliseconds), illustrating consistent performance from both online backdoor detection techniques.

Further performance data can be found in Appendix A.2. We have observed that TESDA achieves up to more than 10x improvement in clock cycle overhead and sub-1% overhead in terms of FLOPS and SIMD operations compared to the baseline.

## 7 Discussion and Future Work

### 7.1 Deployment Challenges in Real-World Scenarios

*Deployment on GPU or Accelerator Devices:* Figure 5 shows that for CPU operation using off-the-shelf libraries such as Scikit-Learn, TESDA incurs comparatively low wall-clock latency compared to the baseline. However, the use of CPU-based libraries and off-the-shelf functions leaves room for parallelization and optimization. A deployment of TESDA on edge GPUs will require building GPU-compatible libraries for outlier detection, parallelized dimension reduction and PCA.

*Deployment in Server Environments:* Tensor parallelism and multi-device pipelining (extensively used in modern deep learning frameworks such as MegatronLM (Shoeybi et al., 2020)) introduces the need for communication and sharing of deep feature values for dimension reduction and PCA. For tensor-parallel workloads TESDA would incur an additional all-reduce operation for transforming deep features (entropy, variance; as discussed in Section 4.1), which costs both memory and computation latency. We therefore concentrate on lightweight edge use cases where single-device deployments without such pipelined parallelism are the norm.

*Deployment to CPU Inference Frameworks:* TESDA has been run on CPU to produce latency analyses in Section 6.4, but libraries such as Scikit-Learn may not be effectively parallelized for CPU inference when run alongside PyTorch as we have here. As such, real-world deployment would require building outlier detection, PCA and dimension reduction functionality that is compatible with a runtime library such as XNNPack to ensure CPU acceleration.

### 7.2 False Alarm Rates in Real-World Settings

We note that the false alarm rate is configurable by the user (See Section 4.3.1) and can be adjusted depending on the consequences of false alarms traded off against the need for detection. Applications such as speedtrap cameras (Vig et al., 2022) that are not necessarily safety-critical may trade off lower false alarm rates against lower attack detection. More safety critical applications such as medical robotics (Akay & Hess,

2019) may require near-total detection coverage and therefore accept higher false alarm rates. As such, the developer making use of TESDA may balance the considerations of possibly falsely flagging an inference as an attack (as a false alarm would) against the need for detection of attacks in edge operation (safe, secure operation). For highly safety critical scenarios, a human-in-the-loop monitoring or alert system for verifying whether an inference result flagged as an attack is truly a misclassification or not may be needed.

An important practical consideration therefore is TESDA's behavior when applied to genuinely clean (non-backdoored) models. In such scenarios, the transformed deep feature distributions should remain consistent between training and test data. The contamination parameter $\epsilon$ controls the robustness of the distribution fitting process, with smaller values leading to more conservative thresholds. While $\epsilon$ does not directly translate to false positive rates, our theoretical framework in Section 4.3 provides bounds on outlier detection behavior. In practice, conservative settings ($\epsilon = 0.01$-$0.02$) typically result in low false alarm rates suitable for precautionary deployment, even when the model's backdoor status is uncertain. Users should monitor operational false positive rates and adjust $\epsilon$ accordingly, as real-world data distributions may experience natural drift or environmental changes that could affect baseline performance.

The ethical considerations of false alarms is likewise something that can be adjusted for by the developer. We also note that TESDA, as a system that flags 'outlier' or uncharacteristic behavior as an attack, may require secondary vetting of classifier outputs if follow-on consequences such as legal action are present for the end user of the CNN application integrated with TESDA. Similar considerations apply to other statistical detection systems for Trojan attacks that rely on on-line detection (e.g. STRIP or TeCo) (Liu et al., 2023).

### 7.3 Addressing the Need for Clean Training Data

TESDA in its current form relies on clean training data to collect distributions of reduced-dimension deep feature PCA coefficients at each layer under examination (as in Section 4). However, Section 6.2 illustrates that TESDA does not require access to the entire training dataset. It has been observed in prior work on Trojan detectors based on activation clustering Chen et al. (2018) that data from different classes produces clusters in the deep feature space of the network. As such, we require sample data or distributions of reduced-dimension deep features characteristic of each class to fit TESDA.

Methods such as the Public Out-of-Distribution Data (POOD) used for constructing the Narcissus backdoor attack (Zeng et al., 2023) may thus be used to build limited distributions of reduced-dimension deep features for each class to fit TESDA. These methods assume that the defender has knowledge of each class label of the network. Synthetic data for each class may also be generated as in (Yuan et al., 2024b) to supplement existing POOD data or de novo through generative learning methods, aiming to minimize classification entropy of generated inputs for each class while subject to stochastic perturbations. Class boundaries or cluster boundaries have been explored using adversarial perturbations in prior art Chakraborty et al. (2021), and such methods may also be applied to enhance a synthetic or POOD dataset. The aim of these methods would be to produce a training-dataset-independent collection of reduced-dimension deep feature PCA coefficients to adequately model nominal CNN behavior.

### 7.4 Deep Feature Dimension Reduction Design Choices

We have chosen to use PCA rather than potentially more powerful nonlinear learning-driven methods such as autoencoders due to several factors, briefly mentioned in Section 4.2:

*Overhead and Computational Efficiency:* PCA allows us to compute a closed-form transformation to project high-dimensional deep features onto an orthobasis, minimizing compute overhead and avoiding the need for tuning or training. In contrast, learning-driven methods such as autoencoders require additional pretraining, a larger volume of data, and careful hyperparameter tuning. Similarly, methods for deep feature selection often rely on heuristic procedures that may not guarantee an optimal lower-dimensional representation, overhead without providing clear benefits in our context. This computational efficiency and linearity is of particular benefit to our use case, for attack detection in edge-based lightweight CNNs.

*Theoretical Optimality and Interpretability:* Under the criterion of minimizing the reconstruction error, PCA produces the optimal linear subspace in terms of variance retention. This is critical for our application, as

backdoor attacks typically induce detectable shifts in the distribution of deep feature activations – shifts that are effectively captured by the principal components. Conversely, while learning-based or heuristic methods are powerful, their non-linear nature complicates the derivation of such optimality guarantees. Prior work such as (Cantareira et al., 2021; Zheng et al., 2022a) has illustrated the effectiveness of simple statistical analysis of deep features or the use of methods such as channel Lipschitzness to derive attack-relevant insights, and highlight the sufficiency of the PCA for this application despite its simplicity.

## 7.5 Limitations of Theoretical Bounds

While Section 4.3 provides principled guidance for setting the contamination parameter $\epsilon$, we acknowledge that the empirical performance sometimes exceeds our theoretical predictions. This discrepancy arises from several factors: (1) our bounds assume perfect multivariate Gaussian distributions, while actual transformed deep feature distributions may exhibit favorable properties for outlier detection that are not captured by worst-case Gaussian analysis; (2) backdoor attacks often induce larger and more systematic distribution shifts than our conservative theoretical framework accounts for; and (3) the MCD estimator demonstrates practical robustness beyond what multivariate Chebyshev inequalities guarantee.

These observations suggest that our theoretical bounds serve as conservative guidelines rather than tight predictions. Future theoretical work could develop more refined bounds that account for the specific characteristics of deep feature distributions and the particular ways backdoor attacks manifest in the transformed feature space. Despite this limitation, the theoretical framework remains valuable for providing interpretable, worst-case guarantees that enable principled hyperparameter selection in safety-critical applications.

## 7.6 Future Work

### 7.6.1 Detector Parameter Space Exploration

Future work in enhancing TESDA envisions further exploration of its design parameters and development of principled methods for building transform functions and dimension reduction methods. Parameterized functions that use adversarial inputs as a test stimulus to optimize the function parameters for maximum true positive rate and minimal false positive rate (in a manner similar to electronic test stimulus optimization (Komarraju et al., 2023)) are one avenue to explore. Adversarial inputs as a means of building optimal detector designs has been explored in (Xiang et al., 2020) and can potentially be applied to TESDA. Future work for training TESDA's outlier detector also envisions exploiting methods discussed in Section 7.3 to estimate the characteristic distribution for reduced-dimension transformed deep feature PCA coefficients without the need for the original clean training dataset (in whole or in part).

Given the wide range of configurations (transforms, PCA coefficient choices, layers to reduce) for TESDA, a further angle of future work is studying false alarm rates for different Trojan contamination thresholds for these parameters. At present, we have examined an all-clean test dataset (where our false alarm rate is recorded, see Section 5.3), and an all-Trojaned dataset to check detection capability. Partial contamination during inference to examine the effects of mixed inputs on the pipeline and mimic real world scenarios is one angle of future work. Another is to vary the extent of data poisoning during trigger injection. We have used attacks prepared by the authors in the benchmark BackdoorBench repository (poisoning ratio of 10%), but future work envisions varying the extent of contamination to produce potentially less effective but more stealthy backdoors to test TESDA.

### 7.6.2 Extensions to Transformer Architectures

Transformer-based models utilize a very different feature and inference structure from convolutional neural networks and it remains unclear as to the effectiveness of TESDA for detection of backdoor attacks in such models. We therefore propose to examine this in future work. While some large language models or multimodal transformers such as Gemma (Team et al., 2024) and LLAMA (Touvron et al., 2023) have used convolutional vision encoders for feature extraction, backdoors in transformer based image processing pipelines may not produce significant deviations in behavior in the vision encoder. We also note that trigger insertion mechanisms for Neural Trojans in transformers have made use of attention hijacking (Lyu et al.,

2022) or attention patch-based triggers (Zheng et al., 2023), which leverage the attention mechanisms that are absent in CNNs. Devising effective and theoretically grounded transforms, dimension reduction strategies and outlier detection for backdoor detection in the deep feature layers of transformers remains an open question.

### 7.6.3 White-Box Attack Scenario Development

This work has examined a 'gray-box' attack scenario, wherein the defender is provided a compromised network to fit the detection mechanism (TESDA) to, with no details as to the attack trigger, target class or injection method.

The wide range of configurations for TESDA makes engineering a 'white-box' attack, where the attacker knows the defense mechanism in order to tailor the backdoor to avoid detection, more difficult. For such an attack, an attacker would have to have prior knowledge of TESDA's exact configuration (the transform chosen, the PCA coefficient chosen for outlier detection and the set of layers chosen for dimension reduction and feature extraction) or alter the attack to cover all possilbe configurations. We note that the range of possible TESDA configurations have examined makes this a nontrivial task.

The second major complication introduced in white-box attack generation is in the fact that TESDA is fit to the network after backdoor injection. As such, for a 'white-box' scenario, the attacker would have to keep the requisite TESDA configuration in the loop during the trigger injection optimizer loop to ensure that every fit of the detector is evaded by the attack. This alteration of the optimizer loop may not be straightforward, especially for GAN-generated attacks such as (Nguyen & Tran, 2020) or subtle injection mechanisms such as color channel quantization in (Wang et al., 2022b). As such, in future work we aim to build such a 'white-box' scenario which does account for or minimize deep feature distribution shifts, which are detected by TESDA to flag backdoor attacks.

## 8 Conclusion

This work has presented TESDA, a low-latency online backdoor attack detection mechanism for lightweight, edge-deployed convolutional neural networks that requires access to the deep features of the neural network. TESDA achieves comparable performance to the state of the art, provides theoretical guarantees on false alarm rate and hyperparameter tuning, and makes no assumptions as to backdoor attack trigger placement, type or injection mechanism. TESDA is moreover trigger-agnostic, and depends solely on deep feature statistics over clean training data.

### Acknowledgments

Acknowledgements removed for review

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

| Dataset | Average Detector Clock Cycles (TESDA) (Normalized w.r.t. Nominal Network) | Clock Cycle Standard Deviation (TESDA) (Normalized w.r.t. Nominal Network) | Average Detector Clock Cycles (STRIP) (Normalized w.r.t. Nominal Network) | Clock Cycle Standard Deviation (STRIP) (Normalized w.r.t. Nominal Network) |
|---|---|---|---|---|
| CIFAR100 | 5.18 | 41.57 | 37.95 | 62.44 |
| GTSRB | 4.13 | 33.59 | 43 | 93.57 |
| Tiny Imagenet | 2.95 | 10.26 | 46.96 | 16.05 |

Table 12: Normalized detector clock cycles required (mean and standard deviation) for TESDA and STRIP (comparing the network with TESDA/STRIP against the network alone) on the Tiny Imagenet, GTSRB and CIFAR100 datasets for Preactresnet-18. The normalization is done w.r.t. the nominal network clock cycle count.

| Dataset | Average Detector Clock Cycles (TESDA) (Normalized w.r.t. Nominal Network) | Clock Cycle Standard Deviation (TESDA) (Normalized w.r.t. Nominal Network) | Average Detector Clock Cycles (STRIP) (Normalized w.r.t. Nominal Network) | Clock Cycle Standard Deviation (STRIP) (Normalized w.r.t. Nominal Network) |
|---|---|---|---|---|
| CIFAR100 | 3.13 | 12.67 | 12.12 | 4.36 |
| GTSRB | 2.88 | 9.54 | 12.52 | 7.77 |
| Tiny Imagenet | 2.92 | 13.59 | 26.82 | 23.19 |

Table 13: Normalized detector clock cycles required (mean and standard deviation) for TESDA and STRIP (comparing the network with TESDA/STRIP against the network alone) on the Tiny Imagenet, GTSRB and CIFAR100 datasets for VGG-19-BN. The normalization is done w.r.t. the nominal network clock cycle count.

# A    Appendix

## A.1    Chernoff Style Bound for $\Delta$

The Chernoff bound (Wainwright, 2019) is a tail bound given by the inequality:

$$\mathbf{P}(d^2 \geq t) \leq \exp{-\phi_x^*(t)} \tag{15}$$

where $\phi_x^*$ is defined as $\sup_{\lambda \geq 0}[\lambda t - \phi(\lambda)]$, and $\phi(\lambda)$ is the logarithm of the moment generating function of $d^2$. Comparing Eqn. 15 with $\mathbf{P}(d^2 \geq \Delta^2) \leq \epsilon$ gives us:

$$\Delta^2 = t \tag{16}$$

As well as

$$\epsilon = \exp{\phi_x^*(t)} \implies \ln{\frac{1}{\epsilon}} = \phi_x^*(t) \tag{17}$$

Since $d^2 \sim \chi_{cN_L}^2$, we know that

$$\phi(\lambda) = -\frac{cN_L}{2}\ln(1 - 2\lambda) \tag{18}$$

Subsequently, differentiating $t + \frac{cN_L}{2}\ln(1 - 2\lambda)$ w.r.t. $\lambda$ and setting it to zero gives:

$$\lambda^* = \frac{t - cN_L}{2t} \tag{19}$$

Which when substituted in the definition of $\phi_x^*(t)$ yields

$$\phi_x^*(t) = \frac{1}{2}(t - cN_L + cN_L\ln(\frac{cN_L}{t})) = \ln{\frac{1}{\epsilon}} \tag{20}$$

Substituting Equation 16 into Equation 20 and solving for $\Delta^2$, we get

$$\Delta^2 = -cN_L W(\frac{\epsilon^{\frac{2}{cN_L}}}{e}) \implies \Delta = \sqrt{-cN_L W(\frac{\epsilon^{\frac{2}{cN_L}}}{e})} \tag{21}$$

where $W(.)$ is the Lambert $W$ function (Bronstein et al., 2008).

## A.2    Additional Overhead Results

### A.2.1    Clock Cycle Analysis

Tables 12 and 13 show mean and standard deviation of clock cycle counts (normalized against the CNN without any detector installed) for STRIP and TESDA. As with Section 6.4, these are calculated over 500 inputs to the network.

| Dataset | Average Detector FLOPS (TESDA) (Normalized w.r.t. Nominal Network) | Average FLOPS (STRIP) (Normalized w.r.t. Nominal Network) | Average Detector SIMD/Vector Ops (TESDA) (Normalized w.r.t. Nominal Network) | Average SIMD/Vector Ops (STRIP) (Normalized w.r.t. Nominal Network) |
|---|---|---|---|---|
| CIFAR100 | 1.002 | 89.55 | 1.029 | 90.11 |
| GTSRB | 1.002 | 89.54 | 1.029 | 89.11 |
| Tiny Imagenet | 1.001 | 100 | 1.021 | 99.96 |

Table 14: Normalized mean detector FLOPS and SIMD operations required for TESDA and STRIP (comparing the network with TESDA/STRIP against the network alone) on the Tiny Imagenet, GTSRB and CIFAR100 datasets for Preactresnet-18. The normalization is done w.r.t. the nominal network clock cycle count. Normalized standard deviation magnitude is of the order of $10^{-2}\%$ of the total or less for SIMD operations and FLOPS and therefore omitted.

| Dataset | Average Detector FLOPS (TESDA) (Normalized w.r.t. Nominal Network) | Average FLOPS (STRIP) (Normalized w.r.t. Nominal Network) | Average Detector SIMD/Vector Ops (TESDA) (Normalized w.r.t. Nominal Network) | Average SIMD/Vector Ops (STRIP) (Normalized w.r.t. Nominal Network) |
|---|---|---|---|---|
| CIFAR100 | 1.002 | 68.55 | 1.015 | 68.38 |
| GTSRB | 1.002 | 68.56 | 1.015 | 68.4 |
| Tiny Imagenet | 1.001 | 96.31 | 1.008 | 96.2 |

Table 15: Normalized mean detector FLOPS and SIMD operations required for TESDA and STRIP (comparing the network with TESDA/STRIP against the network alone) on the Tiny Imagenet, GTSRB and CIFAR100 datasets for VGG-19-BN. The normalization is done w.r.t. the nominal network clock cycle count. Normalized standard deviation magnitude is of the order of $10^{-2}\%$ of the total or less for SIMD operations and FLOPS and therefore omitted.

We see that TESDA's average clock cycle overhead lowers as dataset complexity increases (input size increase and class count rises) for both VGG and Preactresnet-18. We also see that TESDA has consistently lower cycle count standard deviation in the case of Preactresnet-18, but higher standard deviation in VGG due to the large dense classifier module of VGG-19 requiring large, memory-intensive PCA matrix operations for dimensionality reduction that may cause stall cycles (and therefore increase cycle time variability). In all cases, however, TESDA exhibits relative average cycle counts of 7x (CIFAR100, Preactresnet-18) to more than 10x less (Tiny Imagenet, both networks).

This is primarily due to methods such as STRIP (and more recent work such as TeCo Liu et al. (2023)) relying on repeated network inference to detect shifts in behavior characteristic of an attack. For STRIP, this is repeated inference on perturbed inputs, running the CNN up to 100 times on different perturbed variations of the same input to calculate entropy. TESDA achieves this detection of behavioral change by leveraging deep feature shifts.

### A.2.2 FLOP and SIMD Operation Counts

Tables 14 and 15 show the relative average Floating Point Operations per Second (FLOPS) and SIMD operations normalized relative to the network with no detector installed in the case of TESDA and STRIP, for Preactresnet-18 and VGG-19BN respectively. As with Section 6.4, these are averaged over 500 inputs to the network. The normalized standard deviation magnitude is of the order of $10^{-2}\%$ of the total or less for SIMD operations and FLOPS and therefore omitted.

Here we can see that the relative overhead of TESDA is both constant as the dataset complexity increases and that we achieve sub-1% overhead for FLOPS and SIMD operations. Since STRIP relies on repeated inference, its overhead in terms of FLOPS and SIMD operations is many tens of times that of the network itself. The average FLOPS and SIMD operations for STRIP rise compared to the network for Tiny Imagenet due to the substantially larger images *and* class count (GTSRB has a smaller class count but a larger image input size compared to CIFAR-100).

