# OpenReview forum: "Transform-Enabled Detection of Backdoor Attacks in Deep Neural Networks"
_TMLR — Rejected by TMLR_

### Review · Reviewer_pdKq · 2025-03-20

**Summary Of Contributions:**

The paper introduces TESDA, a backdoor attack detection mechanism for deep neural networks. TESDA leverages statistical distributions of reduced-dimension transformed deep features to rapidly detect backdoor-triggered mispredictions. The authors provide theoretical bounds on false alarm rates and demonstrate state-of-the-art detection capability with low latency across multiple attack types and datasets. The paper also evaluates TESDA’s performance against various baselines and conducts extensive ablations to assess the impact of dataset fraction, network architecture, and detection layers on its effectiveness.

**Audience:**

Yes

**Broader Impact Concerns:**

The paper does not sufficiently address how adversaries could adapt to evade TESDA, which is crucial for real-world deployment. A discussion on ethical considerations regarding the false positives and the implications of incorrect detections on security-critical applications should be included.

**Claims And Evidence:**

No

**Requested Changes:**

1.	Provide a comparison between PCA-based dimension reduction and alternative approaches such as autoencoders or deep feature selection to justify the design choice.
2.	Include empirical runtime comparisons between TESDA and competing methods (e.g., STRIP, NeuronInspect) to substantiate claims of low computational overhead.
3.	Analyze the variability in detection rates observed in CIFAR-100 experiments and discuss potential reasons for lower performance.
4.	Address the assumption of access to clean training data by discussing alternative methods for estimating clean feature distributions in the absence of trusted datasets.
5.	Conduct additional experiments to evaluate TESDA’s robustness against adaptive attacks where an adversary is aware of its detection methodology.
6.	Expand the discussion on deployment challenges, including potential false positives in real-world settings and the feasibility of integrating TESDA into production environments.

**Strengths And Weaknesses:**

Strengths:

1. Proposes a novel, modular, and theoretically grounded approach for online backdoor detection.
2. Achieves state-of-the-art detection coverage with low computational overhead.
3. Provides comprehensive experiments across different datasets, attack types, and baselines.
4. Includes theoretical guarantees on false positive rates, enhancing interpretability and robustness.

Weaknesses:

1. The paper lacks clarity in explaining why PCA-based feature reduction is the best choice over alternative techniques like autoencoders or kernel methods.
2. Some claims regarding computational efficiency are not well-supported with direct empirical evidence on runtime comparisons.
3. Certain experimental results, particularly for CIFAR-100, show inconsistent performance, which is not sufficiently analyzed.
4. The assumption of access to clean training data for fitting the detection model is strong and limits practical applicability in some real-world scenarios.
5. The paper does not fully explore the adversarial adaptiveness of TESDA against adaptive attacks designed to bypass statistical anomaly detection.
6. The broader impact of the approach is not well-discussed, particularly regarding potential deployment challenges and adversarial adaptations.

---

> ### Author Response · Authors · 2025-05-22
> **Addressing Comment/Change 1**
>
> 1.	Comment: Provide a comparison between PCA-based dimension reduction and alternative approaches such as autoencoders or deep feature selection to justify the design choice.
>
> Author Response: Our design choice to employ PCA-based dimensionality reduction is motivated by several key considerations (Discussion has been provided in Section 7.4 and highlighted and elaborated on in Section 4.2):
>
> •	Overhead and Computational Efficiency: PCA by virtue of being a linear method that computes a closed-form solution to project high-dimensional deep features onto an orthonormal basis not only minimizes computational overhead but also avoids the need for iterative training or tuning. In contrast, autoencoders – despite their potential to capture non-linear relationships – require additional pretraining, a larger volume of data, and careful hyperparameter tuning, all of which introduce significant extra computational cost and latency.  Similarly, methods for deep feature selection often rely on heuristic procedures that may not guarantee an optimal lower-dimensional representation, thereby adding to the complexity and overhead without providing clear benefits in our context.
>
> •	Theoretical Optimality and Interpretability: Additionally, we note that PCA offers attractive theoretical guarantees. For example, under the criterion of minimizing the reconstruction error, it produces the optimal linear subspace in terms of variance retention. This is critical for our application, as backdoor attacks typically induce detectable shifts in the distribution of deep feature activations – shifts that are effectively captured by the dominant (or, in our case, particularly the lowest-energy) principal components. Conversely, while autoencoders are powerful, their non-linear nature complicates the derivation of such optimality guarantees, and they may suffer from issues like overfitting if not properly regularized.
>
> •	Empirical Evidence from Prior Work: Prior studies support the use of simple, efficient feature extraction methods. For instance, Cantareira et al. (2021) [1] have shown that even straightforward statistical examinations of deep features can reveal the divergence caused by adversarial or backdoor perturbations. Similarly, the CLP approach (Zheng et al., 2022a) [2] demonstrates that relatively simple analyses – such as examining channel Lipschitzness – yield significant attack-relevant insights with minimal overhead. These findings corroborate our empirical observation that PCA, despite its simplicity, is sufficient for capturing the key characteristics of manipulated deep feature distributions.
>
> •	Suitability for Lightweight Edge Applications: Our focus is on lightweight CNNs which are prominent in edge applications. In such scenarios, the low-latency and low-resource properties of PCA are particularly advantageous. The ease of integration with existing benchmarks further reinforces the suitability of PCA for our experimental setup. While alternative methods like autoencoders or complex feature selection techniques might offer improvements in certain settings, their higher overhead and data requirements would be counterproductive in the context of rapid, online backdoor detection on resource-constrained devices.
>
> [1] Gabriel D Cantareira, Rodrigo F Mello, and Fernando V Paulovich. Explainable adversarial attacks in deep neural networks using activation profiles. arXiv preprint arXiv:2103.10229, 2021
>
> [2] Runkai Zheng, Rongjun Tang, Jianze Li, and Li Liu. Data-free backdoor removal based on channel lipschitzness. In European Conference on Computer Vision, pp. 175–191. Springer, 2022a.

---

> > ### Author Response · Authors · 2025-05-22
> > **Addressing Comment/Change 2**
> >
> > 2.	Comment: Include empirical runtime comparisons between TESDA and competing methods (e.g., STRIP, NeuronInspect) to substantiate claims of low computational overhead.
> >
> > Author Response: We have provided wall-clock runtime comparisons in Section 6.4 (Figure 5), where our approach outperforms the STRIP baseline for on-line detection. We note that methods such as NeuronInspect are offline methods applied to a neural network pre-deployment, and take several hours to inspect a network for backdoors [1]. As such, we have provided comparisons to STRIP, which is an on-line method that runs while the CNN is active, much like TESDA.
> >
> > We have further provided additional data on average FLOPS and SIMD operations and clock cycle overhead in Appendix-B, with a brief description of the results highlighted in Section 6.4.  The average and standard deviation of clock cycle overhead of the CNN with the detector installed (normalized, relative to the clock cycle average for the CNN with no detector installed) is shown in Table 14 and 15 for Preactresnet18 and VGG-19-BN respectively. TESDA is seen to have substantially lower cycle time overhead compared to STRIP in all datasets’ cases, with the large linear classifier module of VGG-19-BN causing high variance in cycle time for TESDA due to its memory-intensive matrix operations for dimensionality reduction.
> >
> > Tables 16 and 17 show average Floating-Point Operations Per Second (FLOPS) and SIMD/Vector operation overhead, against normalized for the network with detector (TESDA or STRIP) against the network without it. TESDA is seen to have sub-1% overhead for operations, indicating that parallelization and deployment optimizations may allow substantial improvements in latency and clock cycle time. However, STRIP incurs several tens of times the network’s operation count, limiting the improvements possible for lightweight low-overhead deployment.

---

> > > ### Author Response · Authors · 2025-05-22
> > > **Addressing Comment/Change 3**
> > >
> > > 3.	Comment: Analyze the variability in detection rates observed in CIFAR-100 experiments and discuss potential reasons for lower performance.
> > >
> > > Author Response: We appreciate the reviewer’s insightful observation regarding the variability and lower performance in CIFAR-100 experiments, and have discussed this further in the paper in Section 6. We believe the lower detection performance observed on CIFAR-100 is largely a consequence of its lower resolution, increased noise, and higher class granularity, which collectively yield more variable deep feature representations. We explain the effect of these factors as follows:
> > >
> > > •	Resolution and Noise: CIFAR-100 images are 32×32, which provides significantly less spatial detail compared to datasets like Tiny ImageNet (64×64) or even GTSRB. With fewer pixels per image, the deep features extracted are inherently more sensitive to noise, and even minor perturbations can cause substantial variability in the PCA-based representations. This “noisier” boundary makes it more challenging to distinguish between normal and backdoored behavior reliably.
> > >
> > > •	Class Granularity and Feature Complexity: CIFAR-100 comprises 100 classes, leading to increased intraclass variability and more subtle interclass differences. This fine-grained categorization results in less robust, well-separated feature distributions. As a consequence, the transformation process via PCA – which critically depends on a stable variance structure – faces greater difficulty in establishing a clear detection threshold. In contrast, datasets like Tiny ImageNet and GTSRB provide more distinctive features per image, assisting the outlier detection process.
> > >
> > > •	Training Data Characteristics: The smaller size and higher noise level in CIFAR-100 contribute to a “finicky” detection boundary. The limited and noisier feature information can lead to less accurate estimations of the underlying distribution from the clean training data, thereby affecting the stability of the Mahalanobis distance thresholds used for outlier detection.
> > >
> > > •	Trigger Effects and Signal-to-Noise Ratio: The relative impact of a backdoor trigger on a low-resolution image, such as those in CIFAR-100, may be diminished or confounded by the inherent variability in the dataset. This reduces the signal-to-noise ratio for detecting such manipulations, leading to variability in detection rates.

---

> > > > ### Author Response · Authors · 2025-05-22
> > > > **Addressing Comment/Change 4**
> > > >
> > > > 4.	Comment: Address the assumption of access to clean training data by discussing alternative methods for estimating clean feature distributions in the absence of trusted datasets.
> > > >
> > > > Author Response: We thank the reviewer for raising a promising angle of future work. We have discussed possible methods for addressing the need for clean training data in Section 7.3. TESDA, as an outlier detection method using reduced-dimension transformed deep features, requires a specimen input set that adequately models ‘nominal’ network behavior. The ideal example of this is the full training dataset, and work such as Ranger [1] on error resilience has shown that the extrema of activation statistics of nominal behavior can be modeled with as little as 20% of the clean training dataset (TESDA has tested this in Section 6.2, dataset fraction ablations). In the absence of such trusted data, however, we propose several alternatives in Section 7:
> > > >
> > > > •	Use of Public Out of Distribution Data (POOD) data examples for each class (assuming all classes are known to the defender) may be an option. Such methods have been explored for creation of backdoors in Narcissus [1] to model target distribution behavior, and sufficient amounts of POOD data may be able to model nominal network behavior for reduced-dimension transformed deep features.
> > > >
> > > > •	Synthetic data generation using GANs or methods such as [2] may be explored, minimizing classification entropy of generated inputs while subject to stochastic perturbations to mimic nominal behavior or augment a POOD dataset.
> > > >
> > > > •	A synthetic or POOD dataset may be augmented using adversarial perturbations as surveyed in [2] to find inter-class boundaries, allowing TESDA to be fit to the extrema of nominal but ‘correct’ classifier behavior without the need for trusted training data.
> > > >
> > > > [1] Yi Zeng, Minzhou Pan, Hoang Anh Just, Lingjuan Lyu, Meikang Qiu, and Ruoxi Jia. 2023. Narcissus: A Practical Clean-Label Backdoor Attack with Limited Information. In Proceedings of the 2023 ACM SIGSAC Conference on Computer and Communications Security (CCS '23). Association for Computing Machinery, New York, NY, USA, 771–785. https://doi.org/10.1145/3576915.3616617
> > > >
> > > > [2] Jianhao Yuan, Jie Zhang, Shuyang Sun, Philip Torr, and Bo Zhao. Real-fake: Effective training data synthesis through distribution matching, 2024b. URL https://arxiv.org/abs/2310.10402.
> > > >
> > > > [3] Anirban Chakraborty, Manaar Alam, Vishal Dey, Anupam Chattopadhyay, and Debdeep Mukhopadhyay. A survey on adversarial attacks and defences. CAAI Transactions on Intelligence Technology, 6(1):25–45, 2021

---

> > > > > ### Author Response · Authors · 2025-05-22
> > > > > **Addressing Comment/Change 5**
> > > > >
> > > > > 5.	Comment: Conduct additional experiments to evaluate TESDA’s robustness against adaptive attacks where an adversary is aware of its detection methodology.
> > > > >
> > > > > Author Response: We have examined test cases where the neural network has already been contaminated by a backdoor attack prior to collecting reduced-dimension intermediate layer features and fitting the outlier detector for TESDA, which is not necessarily a black-box but also not a full white-box scenario, as the reviewer has pointed out. To make a white-box scenario for backdoor attacks, an attacker would have to have prior knowledge of TESDA’s exact configuration (the transform chosen, the PCA coefficient chosen for outlier detection and the set of layers chosen for dimension reduction and feature extraction), and also have to alter the attack substantially to evade the configuration in question. We note that the range of possible TESDA configurations we have examined makes this a nontrivial task. This modification, for complex backdoor attacks such as the Input-Aware Backdoor, BPP or WaNet that we have examined, would require substantial additional work and would require the attacker to have TESDA in the loop during optimization to ensure that every fit of the detector is evaded by the attack.
> > > > >
> > > > > As such, we add a discussion on purely white-box scenarios but we note that this would prove to be an entirely new piece of work due to the number of possible TESDA configurations and the work required to tweak a backdoor injection optimizer (which often uses a GAN) to fool a TESDA system that fits itself to the compromised network itself. Due to the complexity of modifying a state of the art attack for full white-box coverage and the number of configurations possible in the modular TESDA approach, we have left white-box examinations of TESDA for future work and have included a discussion of the complexities of white-box examinations of TESDA in the paper in the Discussion section of Section 7.

---

> ### Author Response · Authors · 2025-05-22
> **Addressing Comment/Change 6**
>
> 6.	Comment: Expand the discussion on deployment challenges, including potential false positives in real-world settings and the feasibility of integrating TESDA into production environments.
>
> Author Response: This is a multifaceted topic and will be examined in several steps below. Edits have been provided in the draft for each segment, and mentioned in each segment below.
>
> 6a – Deployment Challenges
>
> A discussion for this has been provided in Section 7.1 and summarized below:
>
> •	Deployment on GPU or Accelerator Devices: TESDA has at present been tested on CPU and using off the shelf libraries such as Scikit-Learn. GPU or accelerator deployments would require building libraries for parallelizing TESDA’s transform and dimensionality reduction to run in parallel to the network, so that the only ‘serial’ components of TESDA are the final layer’s transform and PCA, and the outlier detector.
>
> •	Deployment on CPU: Libraries such as Scikit-Learn are unsuitable for production CPU deployments of CNNs on edge due to overhead and latency. CPU deployments of TESDA would require building libraries in to compile TESDA down to runtime frameworks such as XNNPack for fast CPU inference.
>
> •	Deployment on Server: Server inference on multiple devices introduces the problem of cross-device communication. TESDA requires access to deep features to transform, and its ability to perform outlier detection on reduced-dimension transformed subsets of deep features is uncertain. As such, tensor-parallel environments where each layer is sharded across multiple devices (e.g. MegatronLM) are unsuitable for TESDA. We have also not examined the effects of pipelined inference with sequential layers on multiple devices. TESDA may add latency and overhead to these systems due to communication requirements.
>
> 6b – False Alarm Rates
>
> A discussion of this has been provided in the first paragraph of Section 7.2.
>
> •	We note that the false alarm rate is configurable by the user, with theoretical examination of hyperparameter tuning to match the target False Positive Rate provided in Section 4.3.1. As such, this allows balancing the ethical consequences of flagging an attack with that of safety and security (of high priority in safety or security-critical edge AI systems). Systems such as traffic cameras in speedtraps may not require extreme emphasis on security, while systems such as medical robots may require higher levels of security. A system with high safety requirements but a high penalty for false alarms may also require a human in the loop to check that a flagged outlier is indeed a misclassification, and therefore to reject the inference result (whether or not an attack).
>
> 6c – Ethical Considerations of False Alarm Rates
>
> A discussion of this has been provided in Section 7.2, second paragraph. We note that the adjustable false alarm rate presents an opportunity for the developer here as well. In applications that are not safety critical or where the consequences – legal, personal or otherwise – of a false alarm outweigh the need for detection, the developer may set the outlier detector threshold to minimize false alarms. In the event that the false alarm rate must be as low as possible while maximizing the detection, flagged inputs may be stored or vetted by experts to ensure an attack has indeed occurred before referring the matter for followup legal or investigative action.

---

> > ### Comment · Reviewer_pdKq · 2025-05-28
> > **Response to author rebuttal and revised version**
> >
> > The effort made by the authors to address the reviewers' comments is appreciated. The modifications improve the paper. In the revised version, there are formatting issues that must be revised. E.g., the caption of Fig. 2 is cut.

---

### Review · Reviewer_J3UF · 2025-03-25

**Summary Of Contributions:**

The paper proposes Transform-Enabled Detection of Attacks (TESDA), a practical backdoor detection method in deep neural networks (DNNs). It is training-free, low-latency, and suitable for online deployment. The core idea is that the backdoor triggers cause subtle but statistically detectable shifts in the model’s intermediate feature activations. TESDA captures these shifts by applying a set of reduced-dimension transformations (e.g., feature map variance, entropy, and PCA) to internal layer outputs during the inference. It then uses robust statistical modeling via Minimum Covariance Determinant (MCD) to detect backdoor’s abnormal deviations from the clean data distribution.  The paper verifies that TESDA is able to achieve state-of-the-art detection with very low latency on a variety of attacks, datasets and network backbones.


The contributions include:

1.TESDA implements the reduced-dimension transformation techniques (e.g., feature map variance, entropy, PCA) to model’s intermediate representation. It helps to highlight abnormal patterns induced by the backdoor triggers.

2.The proposed detection method is low-latency and without the clean data label access. It is practical.

3.The detection performance looks good: high true positive rates and low false positive rates.

**Audience:**

No

**Claims And Evidence:**

No

**Requested Changes:**

Please refer to the suggestions in weaknesses.

**Strengths And Weaknesses:**

Strengths:

1.The analysis with intermediate representations and outlier detection is intuitive.

2.The experiment result look good.

3.Please check the contributions section.





Weaknesses:

1.In Figure 1 (Step 1), the method requires extracting representations from all layers of the model, which can introduce computational overhead—especially for deeper or non-standard architectures. Additionally, different model types (e.g., CNNs vs. Transformers) have varied layer designs, which may affect TESDA’s applicability and runtime efficiency. In experiment, the paper only uses similar scale CNN architectures. What about the CNNs are deeper. What’s more, What about transformer-based model architectures? A discussion and further experiments on the scalability of the method across architectures would strengthen the paper.



2.TESDA relies on observing abnormal feature activations caused by backdoor triggers. However, in real-world scenarios, the trigger is unknown. Figure 4 highlights clear differences between clean and triggered samples, but it remains unclear how TESDA ensures trigger activation during detection. Clarifying how the method operates without knowledge of or access to the trigger is crucial for practical deployment. This is very important and please clarify.



3.The paper appears to focus on relatively simple backdoor triggers (e.g., static patterns or small patches). However, modern attacks often use more complex or stealthy triggers (e.g., multiple triggers, global style-based triggers, invisible perturbations), which may not induce strong or localized activation shifts. The evaluation should include such advanced backdoor triggers to better assess TESDA’s robustness.


4.The model architecture only contains light-weighted and simple architectures (e.g., Preactresnet-18 and VGG-19BN). In order to show the generalization ability, there should be more CNN based architectures. Meanwhile, as the Vision transfoirmer is very popular, the experiments should also involve different transformer-based architectures (ViT or CLIP).



5.The backdoor attack baselines used in the experiments are primarily from pre-2021 literature. Recent works propose more stealthy, adaptive, or robust backdoor mechanisms that may not exhibit obvious activation-level anomalies. Including stronger, modern baselines would provide a more rigorous evaluation of the method’s capabilities and limitations.

---

> ### Author Response · Authors · 2025-05-22
> **Addressing Comment 1**
>
> 1.	Comment: In Figure 1 (Step 1), the method requires extracting representations from all layers of the model, which can introduce computational overhead—especially for deeper or non-standard architectures. Additionally, different model types (e.g., CNNs vs. Transformers) have varied layer designs, which may affect TESDA’s applicability and runtime efficiency. In experiment, the paper only uses similar scale CNN architectures. What about the CNNs are deeper. What’s more, What about transformer-based model architectures? A discussion and further experiments on the scalability of the method across architectures would strengthen the paper.
>
> Author Response: We appreciate the reviewer’s concern regarding the computational overhead introduced by extracting representations from all layers, as illustrated in Figure 1 (Step 1), and the question of applicability across different model families. We note the following:
>
> •	Partial-Layer Ablations and Tradeoffs: Our approach is designed to balance detection performance with computational efficiency via partial-layer ablations. In practice, one does not need to extract features from every single layer. In fact, our experimental evaluation includes ablations where only selective layers are used—such as those at the ends of residual blocks or from key dense layers—to ensure that the inherent tradeoff between overhead and detectability is optimized (Section 6.1 illustrates the tradeoff between use of all layers, while we have discussed the configuration using selective layers in Section 5.3). This design choice mitigates the burden of examining every layer in deeper models.
>
> •	Applicability to CNN Architectures: Our experiments have focused on lightweight CNN models (e.g., PreactResNet-18 and VGG-19BN) that are commonly used in edge applications, paralleling those employed in the prior art [1-4]. These architectures share several common characteristics such as convolutional feature extraction, residual/dense connection blocks, and batch normalization layers – all of which allow our method to effectively capture and process the shifts in deep feature distributions. We have provided modular code compatible with the BackdoorBench library so that other researchers may easily integrate and evaluate TESDA on additional architectures within this family.
>
> [1] Yansong Gao, Change Xu, Derui Wang, Shiping Chen, Damith C Ranasinghe, and Surya Nepal. Strip: A defence against trojan attacks on deep neural networks. In Proceedings of the 35th annual computer security applications conference, pp. 113–125, 2019.
>
> [2] Yingqi Liu, et al. Trojaning attack on neural networks. In 25th Annual Network And Distributed System Security Symposium (NDSS 2018). Internet Soc, 2018b.
>
> [3] Runkai Zheng, et al, Data-free backdoor removal based on channel lipschitzness. In European Conference on Computer Vision, pp. 175–191. Springer, 2022a.
>
> [4] Runkai Zheng, et al. Pre-activation distributions expose backdoor neurons. Advances in Neural Information Processing Systems, 35:18667–18680, 2022b.
>
> •	On Scalability and Transformer-Based Models: While our current work is centered on CNN-based architectures, we acknowledge that transformer-based models, which exhibit different layer designs and representation dynamics, may require tailored modifications to the TESDA framework. Transformers have attention mechanisms and different intermediate representations that may not be as amenable to our current low-overhead, transform-enabled detection without further adjustment. We have revised the manuscript to clarify that our method is presently tailored to CNNs and to discuss potential avenues for extending the framework to transformer models in future work.
>
> •	Benchmark Effectiveness and Generalizability: Our work focuses on lightweight CNN architectures which are emblematic of the types of models commonly deployed in edge applications and share fundamental design elements that are prevalent across many lightweight image models. Consequently, the insights we derive from our experiments, particularly regarding the tradeoff between detection effectiveness and computational overhead, are broadly generalizable within this domain. Moreover, our evaluation is based on benchmarks provided by the BackdoorBench repository, which is widely recognized in the literature for its rigorous assessment of backdoor detection methods under realistic constraints. This benchmarking approach, coupled with our ablation studies that optimize layer selection reinforces that our method is effective across representative lightweight CNN models. We stress that our intent is to rigorously validate TESDA within a commonly used framework and that our findings are well-aligned with standard practice in the field. While our current experiments are confined to these benchmark architectures to maintain a clear focus, we view our work as laying a strong foundation.

---

> > ### Author Response · Authors · 2025-05-22
> > **Addressing Concern 2**
> >
> > 2.	Comment: TESDA relies on observing abnormal feature activations caused by backdoor triggers. However, in real-world scenarios, the trigger is unknown. Figure 4 highlights clear differences between clean and triggered samples, but it remains unclear how TESDA ensures trigger activation during detection. Clarifying how the method operates without knowledge of or access to the trigger is crucial for practical deployment. This is very important and please clarify.
> >
> > Author Response: We agree that in a real-world scenario, the trigger is unknown and all that we have access to prior to deployment is the ‘clean’ behavior of a potentially backdoored network. As such, we have noted in Section 3 that TESDA functions based on the distribution of PCA components over clean training data for the network, after the network has been trained (and potentially, backdoored). Our detector does not make any prior assumptions on triggers and simply detects backdoor attacks or behavior indicative of an attack based on the deviation in the intermediate feature PCA component distribution caused by trigger activation (We have highlighted this in the assumptions made in Section 2.3). In the absence of trigger activation, TESDA’s low (and configurable) false alarm rate and theoretical guarantees (Section 4.3) ensure that benign samples are not falsely flagged. In this way, TESDA, like methods such as STRIP, does not rely on assumptions about the backdoor trigger and simply detects attacks when a trigger has been inserted into the input sample (activated). Our only assumption is access to clean training data prior to deployment, and as we show in Section 6.2 (Full ablation in Tables 6-8), we do not require the entire training dataset to fit the outlier detector. This has been discussed (highlighted) in the revised draft in Section 3 and in Figure 4, where we also present diagrams of the use of clean training data distributions for outlier detector training and Trojan attack detection.

---

> > > ### Author Response · Authors · 2025-05-22
> > > **Addressing Comment 3**
> > >
> > > 3. Comment: The paper appears to focus on relatively simple backdoor triggers (e.g., static patterns or small patches). However, modern attacks often use more complex or stealthy triggers (e.g., multiple triggers, global style-based triggers, invisible perturbations), which may not induce strong or localized activation shifts. The evaluation should include such advanced backdoor triggers to better assess TESDA’s robustness.
> > >
> > > Author Response: We have employed the following non-fixed complex triggers, and have included only one fixed trigger pattern that makes use of an image patch (That of TrojanNN [1]), and have seen highly promising results for the Targeted Bit Trojan bitflip based attack that also uses fixed patch trigger patterns. The non-fixed non-patch triggers we have used are:
> > >
> > > ■	We have used GAN-based input-dependent triggers (Input-Aware backdoor [2]). This makes use of multiple nonlocal perturbations and applies a unique trigger to each image.
> > >
> > > ■	We have used imperceptible warping and color channel quantization based triggers in the form of the WaNet [3] attack and the BPP [4] attack respectively.
> > >
> > > ■	We have used an input-dependent trigger based on single samples that minimizes input perturbation, in the form of the SSBA attack [5].
> > >
> > > A table has been provided in the Section 5.1 to highlight the trigger methods and attack methodologies in greater detail. (The new Table 1). We have also added further clarification of trigger methods and complexity in the list of attack test cases in Section 5.3 (highlighted).  Our trigger choices and attack choices were made to provide coverage over a broad range of trigger mechanisms and trigger types, examining visually imperceptible as well as more perceptible triggers and input-dependent or independent trigger mechanisms. This has been elaborated on further in paper in Section 5.1.
> > >
> > > [1] Yingqi Liu, Shiqing Ma, Yousra Aafer, Wen-Chuan Lee, Juan Zhai, Weihang Wang, and Xiangyu Zhang. Trojaning attack on neural networks. In 25th Annual Network And Distributed System Security Symposium (NDSS 2018). Internet Soc, 2018b.
> > >
> > > [2] Tuan Anh Nguyen and Anh Tran. Input-aware dynamic backdoor attack. Advances in Neural Information Processing Systems, 33:3454–3464, 2020.
> > >
> > > [3] Anh Nguyen and Anh Tran. Wanet–imperceptible warping-based backdoor attack. arXiv preprint, arXiv:2102.10369, 2021.
> > >
> > > [4] Zhenting Wang, Juan Zhai, and Shiqing Ma. Bppattack: Stealthy and efficient trojan attacks against deep neural networks via image quantization and contrastive adversarial learning. In Proceedings of the IEEE/CVF Conference on Computer Vision and Pattern Recognition, pp. 15074-15084, 2022b.
> > >
> > > [5] Yuezun Li, Yiming Li, Baoyuan Wu, Longkang Li, Ran He, and Siwei Lyu. Invisible backdoor attack with sample-specific triggers. In Proceedings of the IEEE/CVF international conference on computer vision, pp. 16463–16472, 2021c

---

> > > > ### Author Response · Authors · 2025-05-22
> > > > **Addressing Comment 4**
> > > >
> > > > 4. Comment: The model architecture only contains light-weighted and simple architectures (e.g., Preactresnet-18 and VGG-19BN). In order to show the generalization ability, there should be more CNN based architectures. Meanwhile, as the Vision transfoirmer is very popular, the experiments should also involve different transformer-based architectures (ViT or CLIP).
> > > >
> > > > Author Response: Our work has principally examined lightweight CNNs that have utility as feature extractors or image classification networks on edge platforms, and claims and work descriptions in Sections 1 and 2 have been edited to reflect this. As such, we have concentrated on the Resnet architecture to examine the effects of residual block connections on our deep feature-based approach, and on the VGG architecture to examine the impact of large dense networks (such as the classifier at the end of VGG-19BN) and deep convolutional networks (We have added discussion in Section 5.1 to this effect). We do acknowledge that other platforms are available, but we have also used these test cases to provide data on well-examined lightweight convolutional networks that have been looked at by our baselines as well. We have therefore scaled back our claims in the paper’s abstract and title to reflect this. Furthermore, we have provided code that can be readily integrated with the BackdoorBench repository for examination on other platforms and larger networks.
> > > >
> > > > Prior art such as the baselines we have examined have also looked principally at convolutional networks and attacks on convolutional networks. Transformer based architectures, while growing in popularity for server-side image processing applications, remain too resource-intensive for low-overhead on-device edge tasks, and we therefore scale back our claims to convolutional networks only and leave transformer-based architectures such as VIT and multimodal networks such as CLIP to future work. Similarly, convolutional networks are commonly used as ‘vision encoders’ to patchify and conduct feature extraction on language model pipelines, and remain relevant in modern scenarios for LLMs such as Gemma and LLAMA and other multimodal transformer models. A discussion to this effect has been added in the revisions to Future Work and Conclusions section (Section 7, highlighted).

---

> > > > > ### Author Response · Authors · 2025-05-22
> > > > > **Addressing Comment 5**
> > > > >
> > > > > 5.  Comment: The backdoor attack baselines used in the experiments are primarily from pre-2021 literature. Recent works propose more stealthy, adaptive, or robust backdoor mechanisms that may not exhibit obvious activation-level anomalies. Including stronger, modern baselines would provide a more rigorous evaluation of the method’s capabilities and limitations.
> > > > >
> > > > > Author Response: We note that we have examined attack methods that have used stealthy, adaptive and robust trigger methods that are input-specific, avoiding a fixed trigger pattern or visually obvious trigger (A brief description is provided in Table 1). While one attack we have examined – TrojanNN – is a fixed patch trigger, we have also examined more recent work such as:
> > > > >
> > > > > ■	BPP (2022): This attack has made use of input-dependent quantizations of color channels to produce imperceptible changes in images and force misprediction. It is therefore stealthy, adaptive to the input (since quantization is dependent on the input color channels) and nonlocal (since color channel quantization is image-wide).
> > > > >
> > > > > ■	WANET (2021): This uses subtle warping transforms to perturb the image and trigger a misprediction. WaNet’s warping transforms are again nonlocal and not visually obvious.
> > > > >
> > > > > ■	 Input-Aware Attack (2021): This uses learned triggers generated by GANs that are input-dependent but visually apparent to a human being, producing unique triggers that are nonlocal (thanks to being GAN-generated and not confined to a patch).
> > > > >
> > > > > We have also examined these attacks to provide ablation coverage over a range of different trigger mechanisms, trigger placement, trigger stealth and trigger dependence on input (adaptiveness). These attacks were also examined by benchmark repositories such as BackdoorBench upon the writing of this paper, ensuring that the code we provide and the data we provide can easily be compared to other state of the art methods and examined in future work.  This has been discussed further for clarity in Section 5.1.

---

> > > > > > ### Comment · Reviewer_J3UF · 2025-06-26
> > > > > > **Response to author rebuttal**
> > > > > >
> > > > > > Thank you for the authors’ rebuttal. While it addresses some of my points, I remain concerned about the method’s ability to generalize to larger CNNs and transformer architectures, as well as its evaluation against outdated baseline attacks. Additionally, the updated PDF layout on page 7 is poorly organized, leaving an unnecessary blank area above page 8. I believe including further empirical evidence will help to strengthen the manuscript in future versions.

---

### Review · Reviewer_qXjD · 2025-05-14

**Summary Of Contributions:**

This paper introduces an online Trojan example detection algorithm called TESDA, which identifies Trojaned inputs by analyzing the hidden representations within a backdoored neural network. Specifically, the algorithm collects the hidden representations of a clean data subset as processed by the Trojaned network. To reduce computational overhead, these representations are then transformed. Principal Component Analysis (PCA) is performed on the transformed features, and the resulting PCA coefficients are recorded for a set of predetermined layers for each input in the probing dataset. Each input thus yields a PCA coefficient vector. A multivariate Gaussian distribution is then fitted to these vectors using a reserved clean subset. During the detection phase, test inputs are passed through the Trojaned network to obtain their PCA coefficients. An outlier detection procedure, based on the Minimum Covariance Determinant (MCD) method, is then applied to determine whether an input is potentially Trojaned. The authors also offer quantitative guidance on selecting the key hyperparameter $\Delta$, depending on the user's tolerance for false positives or false negatives.

**Audience:**

Yes

**Broader Impact Concerns:**

not relevant

**Claims And Evidence:**

Yes

**Requested Changes:**

As described in the weaknesses section:

(1) To justify the necessity of the proposed method, the authors should demonstrate that fine-tuning fails to fix a Trojaned model using the same amount of clean data as that required by their algorithm.

(2). An ablation study exam the effect of mismatch between false positive / negative target and actual poisonous rate (proportion of Trojan example in the incoming testing dataset).

(3). Set of experiments showing the false positive rate on a clean model

(4). This limitation should be discussed in more detail, and the derivation or refinement of a more practical or stable criterion for setting
 $\Delta$ would be beneficial.

**Strengths And Weaknesses:**

Strengths:

(1) This paper introduces a novel online Trojan example detection method that can flag compromised inputs without the need to train a complex Trojan detector. Notably, the algorithm is model-agnostic and can be applied to any deep neural network (DNN) in an out-of-the-box fashion—without relying on a clean surrogate model or requiring the collection of a large number of victim model samples.

(2) The authors quantitatively provide guidance for selecting the key parameter $\Delta$ in their outlier detection procedure, enabling users to control the precision–recall trade-off based on their specific tolerance for false positives or false negatives.

(3) The proposed method is evaluated across a range of attack scenarios and compared with several established backdoor detection and defense algorithms. Additionally, thorough ablation studies are conducted to validate the effectiveness of each component and support the robustness of the findings.


Weakness:

(1). The proposed algorithm relies on a subset of clean examples to fit a reference distribution of PCA coefficients. However, one of the most straightforward and effective approaches to mitigating backdoor attacks—when a clean subset is available—is to fine-tune the final layers near the output of the suspect model using the clean data.

(2). One notable omission in the ablation study is an analysis of the effects of underestimating or overestimating the poisoning rate. In practice, users often do not know the actual proportion of poisoned inputs in the testing data, yet the detection procedure relies on an assumed upper bound for this ratio. It is important to understand what happens when there is a mismatch between the assumed poisoning rate (used to set detection parameters) and the true rate. Specifically, how do such discrepancies affect the false positive and false negative rates?

(3). Another missing element in the experimental evaluation is the false positive rate when applying the method to a clean model. In real-world scenarios, users are unlikely to deploy a model if they are aware it is Trojaned. There could be a scenario where a cautious user adopt the proposed algorithm while his/her model is actually clean. Therefore, it is critical to evaluate how often the algorithm incorrectly flags clean inputs in a genuinely clean model. A high rejection rate in such cases would raise serious concerns about the practicality of the approach.

(4).  The suggest choice of parameter $\Delta$ is very hard to control false positive / negative under a reasonable range given a pre-determined target of false negative / positive even given large number of sample size, which cannot explain the rather good control of both type of errors in the empirical study.

---

> ### Author Response · Authors · 2025-05-23
> **Addressing Concern 1**
>
> **Comment**: The proposed algorithm relies on a subset of clean examples to fit a reference distribution of PCA coefficients. However, one of the most straightforward and effective approaches to mitigating backdoor attacks—when a clean subset is available—is to fine-tune the final layers near the output of the suspect model using the clean data.
>
> **Author response**: We thank the reviewer for raising this important comparison. We agree that fine-tuning is a relevant baseline approach when clean data is available. However, TESDA addresses fundamentally different deployment requirements than fine-tuning approaches.
>
> We have added clarification to Section 2.3 explaining that TESDA and fine-tuning serve complementary rather than competing purposes. Fine-tuning requires taking the model offline, careful hyperparameter selection, and permanently modifies model weights, potentially degrading performance on edge cases not represented in the limited clean data. In contrast, TESDA provides online protection during inference without modifying the original model.
>
> Critically, fine-tuning addresses backdoors at the model level but cannot detect individual poisoned inputs during deployment, whereas TESDA enables real-time flagging of suspicious inputs for human review or rejection. For safety-critical applications requiring continuous operation, the ability to identify and handle suspicious inputs in real-time while maintaining the original model's performance is essential. We have also strengthened our motivation in Section 1 to clarify that TESDA targets scenarios requiring online detection during inference rather than offline model remediation.

---

> > ### Author Response · Authors · 2025-05-23
> > **Addressing Concern 2**
> >
> > **Comment**: One notable omission in the ablation study is an analysis of the effects of underestimating or overestimating the poisoning rate. In practice, users often do not know the actual proportion of poisoned inputs in the testing data, yet the detection procedure relies on an assumed upper bound for this ratio. It is important to understand what happens when there is a mismatch between the assumed poisoning rate (used to set detection parameters) and the true rate. Specifically, how do such discrepancies affect the false positive and false negative rates?
> >
> > **Author response**: We appreciate the reviewer’s observation regarding practical deployment scenarios. This is an important limitation of our current work that deserves more thorough investigation.
> > The contamination parameter ϵ serves as an upper bound estimate for the proportion of outliers in the training data when fitting the MCD distribution, rather than requiring a precise poisoning rate for incoming test data. Based on our theoretical framework in Section 4.3.1, underestimating ϵ should result in more conservative detection thresholds (potentially higher false negatives but lower false positives), while overestimating ϵ should increase detection sensitivity (potentially higher true positives but also higher false positives).
> >
> > We have added discussion in Section 7.2 acknowledging this limitation and recommending that practitioners start with conservative estimates (ϵ = 0.01-0.02) and monitor operational performance to adjust parameters as needed. However, we acknowledge that a systematic empirical analysis of the robustness to ϵ misestimation would strengthen our work and represents an important direction for future validation of TESDA's practical applicability.

---

> > > ### Author Response · Authors · 2025-05-23
> > > **Addressing Concern 3**
> > >
> > > **Comment**: Another missing element in the experimental evaluation is the false positive rate when applying the method to a clean model. In real-world scenarios, users are unlikely to deploy a model if they are aware it is Trojaned. There could be a scenario where a cautious user adopt the proposed algorithm while his/her model is actually clean. Therefore, it is critical to evaluate how often the algorithm incorrectly flags clean inputs in a genuinely clean model. A high rejection rate in such cases would raise serious concerns about the practicality of the approach.
> > >
> > > **Author response**: This is a critical practical consideration that we have now addressed in Section 7.2. When TESDA is applied to clean (non-backdoored) models, the transformed deep feature distributions should remain consistent between training and test data. While ϵ does not directly translate to false positive rates, in practice, conservative settings typically result in low false alarm rates suitable for precautionary deployment, even when the model's backdoor status is uncertain.
> > >
> > > Users could monitor operational false positive rates and adjust ϵ accordingly, as real-world data distributions may experience natural drift or environmental changes that could affect baseline performance. We believe that the theoretical framework provides principled guidance for threshold selection that would make TESDA safe to deploy even in uncertain scenarios.

---

> > > > ### Author Response · Authors · 2025-05-23
> > > > **Addressing Comment 4**
> > > >
> > > > **Comment**: The suggested choice of parameter Δ is very hard to control false positive / negative under a reasonable range given a pre-determined target of false negative / positive even given large number of sample size, which cannot explain the rather good control of both type of errors in the empirical study.
> > > >
> > > > **Author response**: We acknowledge this limitation in our theoretical analysis and have added a new subsection (7.5) specifically addressing this discrepancy between theoretical bounds and empirical performance, as per the reviewer’s suggestion.
> > > >
> > > > The gap between theory and practice arises from several factors: (1) our bounds assume perfect multivariate Gaussian distributions, while actual transformed deep feature distributions may exhibit favorable properties for outlier detection not captured by worst-case Gaussian analysis; (2) backdoor attacks often induce larger and more systematic distribution shifts than our conservative theoretical framework accounts for; and (3) the MCD estimator demonstrates practical robustness beyond what multivariate Chebyshev inequalities guarantee.
> > > >
> > > > We now clarify that our theoretical bounds serve as conservative guidelines rather than tight predictions. This provides interpretable, worst-case guarantees for safety-critical applications while acknowledging that empirical performance often exceeds these bounds. We identify the development of more refined theoretical bounds that account for deep feature distribution characteristics as important future work.

---

> > > > > ### Comment · Reviewer_qXjD · 2025-06-13
> > > > > **response to author's rebuttal**
> > > > >
> > > > > Thanks for authors' response and effort. While the answer partially addressed my concern, I still suggest to add more empirical or theoretical evidence in your future version to support your conclusion regarding the argument I raised in my discussion.

---

### Author Response · Authors · 2025-05-22
**Note on paper edits**

We thank the reviewers for their comments and feedback. The authors would like to note that per IEEE editing guidelines for journal publications, changes in the draft main body have been highlighted for convenience during the review process. Appendix-A.2 has been added to address reviewer concerns regarding hardware overhead, and a highlighted summary of its findings has been presented in Section 6.4. We have also included an additional section 7.5 that more clearly addresses the limitations of our theoretical bounds.

---

### Decision · Action_Editor_BPRd · 2025-06-29

**Recommendation:** Reject

**Audience:**

Yes

**Audience Explanation:**

The paper addresses detection of backdoor attack to machine learning models. The problem is of considerable interest within the community.

**Claims And Evidence:**

No

**Claims Explanation:**

This paper addresses backdoor detection problem. The core idea is to treat backdoor input as an outlier detection problem. The proposed method applies PCA to latent features and learn a distribution based clean samples. At test time, it test the latent feature of a new input against the distribution and consider out-of-distribution samples as triggered inputs.

A few fundamental issues raised by reviewers are not addressed satisfactory.

1, The practical justification for the proposed approach, particularly when compared to simply fine-tuning the model with clean samples, remains unconvincing. The authors' rebuttal did not sufficiently address this concern.
2, The proposed method's reliance on parameters for PCA and outlier detection is a significant practical concern. These parameters could be difficult to determine consistently across varying poisoning rates, model architectures, and other real-world conditions, potentially hindering its applicability.
3, The baselines used are outdated, and the method's extendability to larger-scale convolutional networks and transformers remains unclear.